# Mindfulness-based programmes for mental health promotion in adults in nonclinical settings: A systematic review and meta-analysis of randomised controlled trials

Julieta Galante [1,2]*, Claire Friedrich[1], Anna F Dawson[3], Marta Modrego-Alarcón[4,5], Pia Gebbing[6], Irene Delgado-Suárez[4,7], Radhika Gupta[1], Lydia Dean[1], Tim Dalgleish[1,8], Ian R White[9], Peter B Jones[1,2,8]

1 University of Cambridge, Cambridge, United Kingdom, 2 National Institute for Health Research Applied Research Collaboration East of England, Cambridge, United Kingdom, 3 Australian National University, Canberra, Australia, 4 University of Zaragoza, Zaragoza, Spain, 5 Primary Care Prevention and Health Promotion Research Network (RedIAPP), Zaragoza, Spain, 6 Leiden University, Leiden, the Netherlands, 7 Institute of Medical Research Aragón, Zaragoza, Spain, 8 Cambridgeshire and Peterborough NHS Foundation Trust, Cambridge, United Kingdom, 9 University College London, London, United Kingdom

* mjg231@cam.ac.uk

**Data Availability Statement:** All data files are available from the University of Cambridge

## Abstract

### Background

There is an urgent need for mental health promotion in nonclinical settings. Mindfulness–based programmes (MBPs) are being widely implemented to reduce stress, but a comprehensive evidence synthesis is lacking. We reviewed trials to assess whether MBPs promote mental health relative to no intervention or comparator interventions.

### Methods and findings

Following a detailed preregistered protocol (PROSPERO CRD42018105213) developed with public and professional stakeholders, 13 databases were searched to August 2020 for randomised controlled trials (RCTs) examining in–person, expert–defined MBPs in nonclinical settings. Two researchers independently selected, extracted, and appraised trials using the Cochrane Risk–of–Bias Tool 2.0. Primary outcomes were psychometrically validated anxiety, depression, psychological distress, and mental well–being questionnaires at 1 to 6 months after programme completion. Multiple testing was performed using $p < 0.0125$ (Bonferroni) for statistical significance. Secondary outcomes, meta–regression and sensitivity analyses were prespecified. Pairwise random–effects multivariate meta–analyses and prediction intervals (PIs) were calculated.

A total of 11,605 participants in 136 trials were included (29 countries, 77% women, age range 18 to 73 years). Compared with no intervention, in most but not all scenarios MBPs improved average anxiety (8 trials; standardised mean difference (SMD) = −0.56; 95% confidence interval (CI) −0.80 to −0.33; $p$–value < 0.001; 95% PI −1.19 to 0.06), depression (14 trials; SMD = −0.53; 95% CI −0.72 to −0.34; $p$–value < 0.001; 95% PI −1.14 to 0.07), distress

Repository (digital object identifier https://doi.org/10.17863/CAM.59644).

**Funding:** This publication presents independent research funded by the National Institute for Health Research (NIHR). The views expressed are those of the authors and not necessarily those of the NHS, the NIHR or the Department of Health and Social Care. JG is funded by a NIHR Post-doctoral Fellowship for this research project (salary and all project costs, PDF-2017-10-018, https://www.nihr.ac.uk/), she is also supported by the NIHR Applied Research Collaboration (ARC) East of England (grant awarded to PBJ, RNAG/564, https://arc-eoe.nihr.ac.uk/). CF's salary for this research project was funded by a Cambridgeshire & Peterborough NHS Foundation Trust grant awarded to JG (RNAG/552, https://www.cpft.nhs.uk/). IW was supported by the UK Medical Research Council (MC_UU_12023/21, https://mrc.ukri.org/). TD was supported by the UK Medical Research Council (SUAG/043 G101400, https://mrc.ukri.org/), the Wellcome Trust (104908/Z/14/Z, 107496/Z/15/Z, https://wellcome.org/), and the NIHR Cambridge Biomedical Research Centre (RG85446, 247730, https://cambridgebrc.nihr.ac.uk/). PBJ is supported by the Wellcome Trust (095844/Z/11/Z, https://wellcome.org/), the UK Medical Research Council (MR/N019067/1, https://mrc.ukri.org/), and the NIHR ARC East of England (RNAG/564, https://arc-eoe.nihr.ac.uk/). MMA and IDS were supported by the Spanish Ministry of Education, Culture and Sport (FPU15/00598 and FPU16/03565 respectively, http://www.mecd.gob.es/). The funders had no role in study design, data collection and analysis, decision to publish, or preparation of the manuscript.

**Competing interests:** The authors have declared that no competing interests exist.

**Abbreviations:** CES–D, Center for Epidemiologic Studies Depression Scale; CI, confidence interval; COVID–19, Coronavirus Disease 2019; GRADE, Grading of Recommendations Assessment, Development and Evaluation; ICC, intracluster correlation coefficient; ICTRP, International Clinical Trials Registry Platform; IPD, individual participant data; MBP, mindfulness–based programme; MBSR, Mindfulness–Based Stress Reduction; PANAS–P, Positive Affect Schedule; PI, prediction interval; PRISMA, Preferred Reporting Items for Systematic Reviews and Meta–Analyses; RCT, randomised controlled trial; SMD, standardised mean difference.

(27 trials; SMD = −0.45; 95% CI −0.58 to −0.31; $p$–value < 0.001; 95% PI −1.04 to 0.14), and well–being (9 trials; SMD = 0.33; 95% CI 0.11 to 0.54; $p$–value = 0.003; 95% PI −0.29 to 0.94). Compared with nonspecific active control conditions, in most but not all scenarios MBPs improved average depression (6 trials; SMD = −0.46; 95% CI −0.81 to −0.10; $p$–value = 0.012, 95% PI −1.57 to 0.66), with no statistically significant evidence for improving anxiety or distress and no reliable data on well–being. Compared with specific active control conditions, there is no statistically significant evidence of MBPs' superiority. Only effects on distress remained when higher–risk trials were excluded. USA–based trials reported smaller effects. MBPs targeted at higher–risk populations had larger effects than universal MBPs. The main limitation of this review is that confidence according to the Grading of Recommendations Assessment, Development and Evaluation (GRADE) approach is moderate to very low, mainly due to inconsistency and high risk of bias in many trials.

## Conclusions

Compared with taking no action, MBPs of the included studies promote mental health in nonclinical settings, but given the heterogeneity between studies, the findings do not support generalisation of MBP effects across every setting. MBPs may have specific effects on some common mental health symptoms. Other preventative interventions may be equally effective. Implementation of MBPs in nonclinical settings should be partnered with thorough research to confirm findings and learn which settings are most likely to benefit.

## Author summary

### Why was this study done?

- Mindfulness courses to increase well–being and reduce stress have become very popular; most are in community settings.

- Many randomised controlled trials (RCTs) tested whether mindfulness courses show benefit, but results are varied and, to our knowledge, there are no reviews combining the data from these studies to show an overall effect.

### What did the researchers do and find?

- Worldwide, we identified 136 RCTs on mindfulness training for mental health promotion in community settings. We reviewed them all, assessed their quality, and calculated their combined effects.

- We showed that, compared with doing nothing, mindfulness reduces anxiety, depression, and stress, and increases well–being, but we cannot be sure that this will happen in every community setting.

- In these RCTs, mindfulness is neither better nor worse than other feel–good practices such as physical exercise, and RCTs in this field tend to be of poor quality, so we cannot be sure that our combined results represent the true effects.

### What do these findings mean?

- Mindfulness courses in the community need to be implemented with care, because we cannot assume that they work for everyone, everywhere.

- We need good quality collaborative research to find out which types of communities benefit from the different types of mindfulness courses available.

- The courses that work best may be those aimed at people who are most stressed or in stressful situations.

## Introduction

With major depression listed as the second largest cause of global years lived with disability, and several other mental disorders within the top 25 [1], there is a widely recognised need to prioritise preventative programmes [2]. Such programmes, introduced across schools, workplaces, and communities, usually target psychological distress which, if unaddressed, can result in mental and physical disorders [3]. The public is willing to take preventative action: 65% would practice something for 15 minutes per day if that could reduce their stress by 30% [4].

Commonly implemented preventative interventions include mindfulness–based programmes (MBPs), which typically define mindfulness as "the awareness that emerges through paying attention on purpose, in the present moment, and nonjudgmentally to the unfolding of experience moment by moment" [5]. Recently, a group of prominent mindfulness teachers have agreed on what MBPs should comprise: sustained training in formal and informal mindfulness meditation, scientific approaches to managing health, suitability for delivery in public institutions across a range of settings and cultures, and class–based experiences of collective and individual inquiry with a qualified teacher in a participatory learning process [6]. The United Kingdom National Health Service offers therapies that are based on mindfulness [7]. However, the cultural traditions from which mindfulness stems do not conceptualise it as a therapy [8]; this has encouraged advocates, first in the United States of America (USA) and thereafter more globally, to widely promote nontherapeutic secular mindfulness training [9]. Currently, in the USA, twice as many people practice mindfulness for wellness than for treating a specific health condition [10]. MBPs, ubiquitous in high–income countries, are frequently promoted as the go–to universal tool to reduce stress and increase well–being, accessible to anyone, anywhere [8].

Trials assessing MBPs in nonclinical settings have quickly accumulated in recent years. Systematic reviews have synthesised findings from MBPs from educators [11,12], parents [13], caregivers [14,15], healthcare professionals [16–21], athletes [22], working adults [23–27], older adults [28], university students [29–31], and the general population [32–36], primarily focusing on wellness and mental health outcomes. Most, but by no means all results favour MBPs over comparison conditions. These reviews tend towards broad inclusion criteria, reflecting the state of the science at the time they were conducted. As well as gold standard randomised controlled trials (RCTs), both uncontrolled and nonrandomised trial findings are many times included. In contrast, literature searches often then exclude important areas of grey literature such as unpublished RCTs, as well as studies in languages other than English. Some reviews also combine both nonclinical and clinical MBPs or include other interventions,

and methods are not prespecified. Finally, formal meta–analysis of the synthesised data is infrequent and sometimes (but not always, e.g., [26,37]) neglects to disaggregate trials with active versus passive control groups.

There is now a critical mass of good quality RCTs of MBPs in nonclinical settings. Consequently, in line with calls to improve mindfulness research quality, we conducted a comprehensive systematic review and meta–analysis of MBPs in nonclinical settings targeted at mental health that focused only on RCT data [38,39]. Our primary question was whether these MBPs improve anxiety, depression, psychological distress, and/or well–being 1 to 6 months after programme completion, relative to no intervention or comparator interventions.

## Methods

Our review procedures were developed with public and professional stakeholders [40,41] and published in detail in a prospective protocol [42]. Stakeholders shaped the research questions, prioritised outcomes and moderation analyses, selected studies, extracted data, interpreted results, and developed lay summaries. This study is reported as per the Preferred Reporting Items for Systematic Reviews and Meta–Analyses (PRISMA) guideline (S1 Checklist) [43].

### Study search and selection

Thirteen databases were electronically searched for eligible studies from inception to 1 August 2020: AMED, ASSIA, CENTRAL, CINAHL, ERIC, EThOS, EMBASE, MEDLINE, ProQuest, PsycINFO, Scopus, Web of Science, and the World Health Organization International Clinical Trials Registry Platform (ICTRP). No geographical, language, or publication date/type restrictions were applied (S1 Appendix). Publication references were inspected for further studies. Unpublished reports were included (e.g., theses). Nonpublic sources of studies (e.g., private datasets) were not sought [44], but authors were contacted to seek clarification or data from which effect sizes could be calculated if such data were not available in their report. We searched ICTRP to find further trials and to assess publication bias.

Studies were deemed eligible if they: (1) were parallel–arm RCTs including cluster–RCTs; (2) assessed group–based first–generation MBPs as defined in Crane [6], with a minimum intensity of 4 one–hour in–person teacher–led sessions or equivalent (4 MBP sessions were used as the "minimum dose" for participants in previous studies [45], and 1–hour sessions are common in nonclinical busy settings [46]); (3) included adult (18+ years old) participants living in the community, as long as the trial had not selected them for having any particular clinical condition (MBPs targeting specific community groups were included); (4) reported at least one of the prespecified outcomes of interest (see below); and (5) compared MBPs with a control group (i.e., not just with a different type of eligible MBP). Online MBPs were excluded as we believe they are different enough from in–person MBPs (e.g., typically not group–based, and fully or semiautomated) to merit their own separate analysis [47].

Using Covidence software [48], 2 reviewers independently assessed the titles and abstracts of retrieved records against inclusion criteria. Full texts were obtained for abstracts not deemed irrelevant by both and again independently assessed for eligibility. Multiple reports of the same trial were combined. Two researchers independently extracted the information from the included full–text papers using prepiloted forms (S1 Appendix). Disagreements were discussed and resolved within the review team.

### Outcomes: Organisation, assessment, and transformation

The 4 primary outcome domains were anxiety, depression, psychological distress, and mental well–being, measured in a primary time range of between 1 and 6 months following

programme completion. Measures taken less than 1 month after programme completion may not inform stable changes making them less clinically relevant, so this "post–intervention" time range was considered as a secondary outcome, as was the time range of follow–ups longer than 6 months post–intervention. Other secondary outcome domains included cognitive functioning (assessed using experimental tasks), real life functioning (e.g., professional performance), relationship with the self (e.g., self–esteem, self–compassion), and psychosomatic outcomes (e.g., sleep, pain). Adverse event or effect data were recorded. In view of the high number of trials reporting dispositional mindfulness, we included it as a mechanistic outcome, although it is not in the review protocol. Outcomes deemed not to belong to any of the outcome domains described above were excluded from the review. All self–reported outcomes had to be psychometrically validated in the language used and could not just measure momentary states [49]. If a study measured an outcome more than once within these prespecified time ranges, the longer follow–up was used. When trials reported more than 1 measure of the same outcome within the same time range, or more than 1 eligible sample, we applied the prioritisation criteria set out prospectively in our protocol [42]. For example, we preferred trial primary outcomes and intention–to–treat samples. Trial outcomes were preliminarily categorised into the review outcome domains before analysis via discussion between reviewers extracting the data, with final categorisation made by senior team members blind to trial results and to which trial measures belonged (S1 Appendix).

The standardised mean difference adjusted for small sample bias (SMD, or Hedges' g) was used as a measure of treatment effect [50]. When baseline outcome values were reported, we calculated SMD using the ANCOVA estimate [51]. When missing from trial reports, within–study baseline–endpoint correlations were calculated from publicly available individual participant data (IPD) or imputed as follows. For distress and well–being outcomes, we assumed that within–study baseline–endpoint correlations for each time point were the same as in the IPD from a trial recently conducted by some of us and included in this review [52]. For the other outcomes, we took the mean of the correlations available in other studies. When baseline data were not available, we calculated SMD using adjusted (if available) or unadjusted final values analyses [53–55]. Missing standard deviations were imputed averaging those of other time points within the same study and outcome, or, if not available, from other studies using the same instrument. Subscales were combined when possible using their correlations. Ordinal and categorical data were transformed to be pooled together with continuous data [42]. When outcome sample size was missing, it was estimated from other data. We accounted for clustering when this was missing in cluster–RCT reports [42].

Control groups were grouped into categories following related reviews to facilitate comparison [56,57]: (1) no intervention or wait–list ("passive controls"); (2) interventions designed principally to take account of nonspecific therapeutic factors such as receiving attention from a teacher, without expected specific effects on outcomes of interest ("active nonspecific controls"); and (3) interventions with active ingredients specifically designed to augment change in our outcomes ("active specific controls"). When trials had multiple control groups fitting 1 category, these were combined. In multiarm trials with 2 MBPs, these were combined.

Two researchers independently assessed trials' methodological quality for the included outcomes using the Cochrane revised tool for assessing risk of bias in randomised trials (RoB2, version 9 October 2018) for RCTs and cluster–RCTs [58]. This tool stringently measures potential bias across 5 sources (called "domains" in the tool): (1) randomisation; (2) deviations from intended interventions; (3) missing outcome data; (4) measurement of the outcome; and (5) selection of the reported result. None of the authors assessed risk of bias of their own trial. When data were unavailable for outcomes mentioned in trials' public registers or publications, this was interpreted as known nonreporting bias. Potentially eligible trial registry records with

no published results were considered suggestive of nonreporting bias. We attempted to contact authors if trial enrolment started more than 3 years before our search date and deemed a trial as unpublished if authors offered either no outcome reports or an account of their absence [59]. Small–study effects suggesting unknown nonreporting and other biases were assessed by visual inspection of funnel plots in meta–analyses of primary outcome domains with at least 10 studies. We used the Grading of Recommendations Assessment, Development and Evaluation (GRADE) approach to assess confidence in the cumulative evidence [60]. It classifies the quality of evidence for each result in 1 of 4 levels of certainty—high, moderate, low, and very low. For each primary outcome we considered trials' risk of bias, meta–analysis nonreporting bias, imprecision (confidence intervals (CIs)) inconsistency (prediction intervals (PIs)), and indirectness of evidence.

## Data synthesis

We used Stata/SE 16.1 [61] to compute pairwise random–effects meta–analyses within comparator categories and applied a conservative Bonferroni correction for multiple testing to each of the 4 primary outcome domains using $p < 0.0125$ as the critical level for significance to maintain the overall type I error rate at 0.05 [62]. We included the 4 primary outcome domains in a multivariate meta–analysis using all the prespecified time point ranges available (i.e., post–intervention, 1 to 6 months later, over 6 months later). Multivariate meta–analysis differs from univariate meta–analysis in that it takes into account within–and between–study correlations, reducing bias, and improving precision [63]. Stata's mvmeta package was employed [64,65]. Within–study correlations between outcome domains were estimated from our IPD and assumed to apply to the other studies [52]. Between–study variance–covariance matrices were estimated as unstructured using restricted maximum likelihood; if not possible, they were estimated as exchangeable with the fixed correlation that yielded the largest restricted log likelihood.

Multivariate meta–analyses for secondary outcome domains also included all available time point ranges, but data from our IPD were less suitable to estimate between–outcome within–study correlations and no IPD were available, so meta–analyses were outcome–specific. Within–study correlations were bypassed using Riley's method as our IPD were unsuitable [66]; for cognitive functioning, this method had to be rejected due to extreme correlations, so a within–study correlation of 0.75 was imputed (high given the outcome–specific analysis) with a sensitivity analysis testing 0.5 [67]. Results of meta–analyses containing few studies were interpreted cautiously, including multivariate meta–analyses with outcomes derived from a single trial. When multivariate meta–analyses failed to converge, results of univariate meta–analyses were reported. As a measure of real–life implications of between–study heterogeneity, prediction intervals were estimated reflecting the variation in intervention effects over the different trial settings [68,69].

We conducted prespecified sensitivity analyses on primary outcome domains where we had data from at least 10 studies. These explored sensitivity of results to (1) overall and bias–source–specific risks of bias, by removing trials with higher risk of bias; (2) within–study correlation assumptions, by using Riley's estimation method, and by conducting univariate meta–analyses; (3) standard deviation imputations, by inflating them by 10%; (4) imputing intracluster correlation coefficients (ICCs), by using ICC = 0.10; and (5) skewed data, by excluding data coming from samples of fewer than 30. Post hoc sensitivity analyses were also conducted as outlined in the Results.

We conducted prespecified moderator analyses on primary outcome domains of the following study–level characteristics: (1) region, comparing trials from the USA—where MBPs are

most established in nonclinical settings—with the rest of the world; (2) type of participant, grouping interventions into universal (for anyone), selective (for those at higher risk of developing mental health problems, such as carers), or indicated (for individuals with subclinical symptoms of mental health conditions) [70]; (3) intervention duration; (4) additional inclusion of intervention components other than the activities common to all MBPs [6]; and (5) active control type. We conducted these only when there were at least 10 studies with moderator information. We used random–effects multivariable meta–regression within multivariate meta–analyses and interpreted with caution analyses of categorical subgroup variables with fewer than 5 studies per category [71].

## Results

A study selection flowchart is shown in Fig 1. Too much information was missing to assess eligibility when only conference abstracts were available from databases or authors (S1 Appendix). MBP teachers are required to be well trained, but many trial reports do not describe their credentials [6]. We (1) included these studies, excluding only those explicitly mentioning insufficient training; and (2) conducted an ad hoc sensitivity analysis only including studies which suggested criteria–concordant training (97 trials, 71%) to see if results would differ.

### Study characteristics

One hundred and thirty–six trials were eligible for meta–analysis, 129 participant–level RCTs, and 7 cluster–RCTs. Table 1 summarises the main characteristics of the included studies. Trials were conducted between 1997 and 2020 across 29 countries. Almost half of the trials were completed in North America (mainly USA), 37 in Europe, 19 in Asia (mainly China), 6 in the Middle East, 5 in Australia, and 4 in South America. Sample sizes varied from 18 to 616 participants with a median of 60. Mean ages ranged from 18 to 73 years old, and the gender balance differed between trials with a mean of 77% women. Sixteen trials (12%) recruited stressed individuals for whom the MBP was considered an indicated preventative intervention. MBPs were selective interventions in 47 trials (35%), targeting groups such as healthcare workers, medical interns, carers, school teachers, and pregnant women. The remaining 73 trials (54%) used "universal" self–selected samples like community adults, students, employees, or older adults. Those with severe mental health problems or recent stressful life events were excluded in 99 trials (73%).

MBPs were optional courses in all of the settings. The most common MBP was Mindfulness–Based Stress Reduction (MBSR) [5], sometimes slightly adapted, assessed in 44 (32%) trials. The most common additional component was physical activity (60 studies, 44%). MBP group sizes ranged from 6 to 30 participants per group. Planned intervention contact hours ranged from 4 to 30 hours with a mean of 16. Information about MBP teachers typically lacked detail (e.g., teacher background).

Trials measured a wide range of outcomes within our domains of interest. The most common primary outcome measures were: for anxiety the Beck Anxiety Inventory, for depression the Center for Epidemiologic Studies Depression Scale (CES–D), for psychological distress the Perceived Stress Scale, and for mental well–being the Positive Affect Schedule (PANAS–P). Psychological distress was the most commonly measured outcome domain (102 trials, 75%). All of the outcome measures were self–reported except for some real–life functioning outcomes such as exams, some psychosomatic outcomes such as peri–labour opioid use, and all cognitive functioning outcomes which involved experimental tasks. Follow–up times ranged from post–intervention (most trials) to an outlier of 6 years [72]. The most common control group was passive (no intervention or waitlist), used in 96 trials (71%).

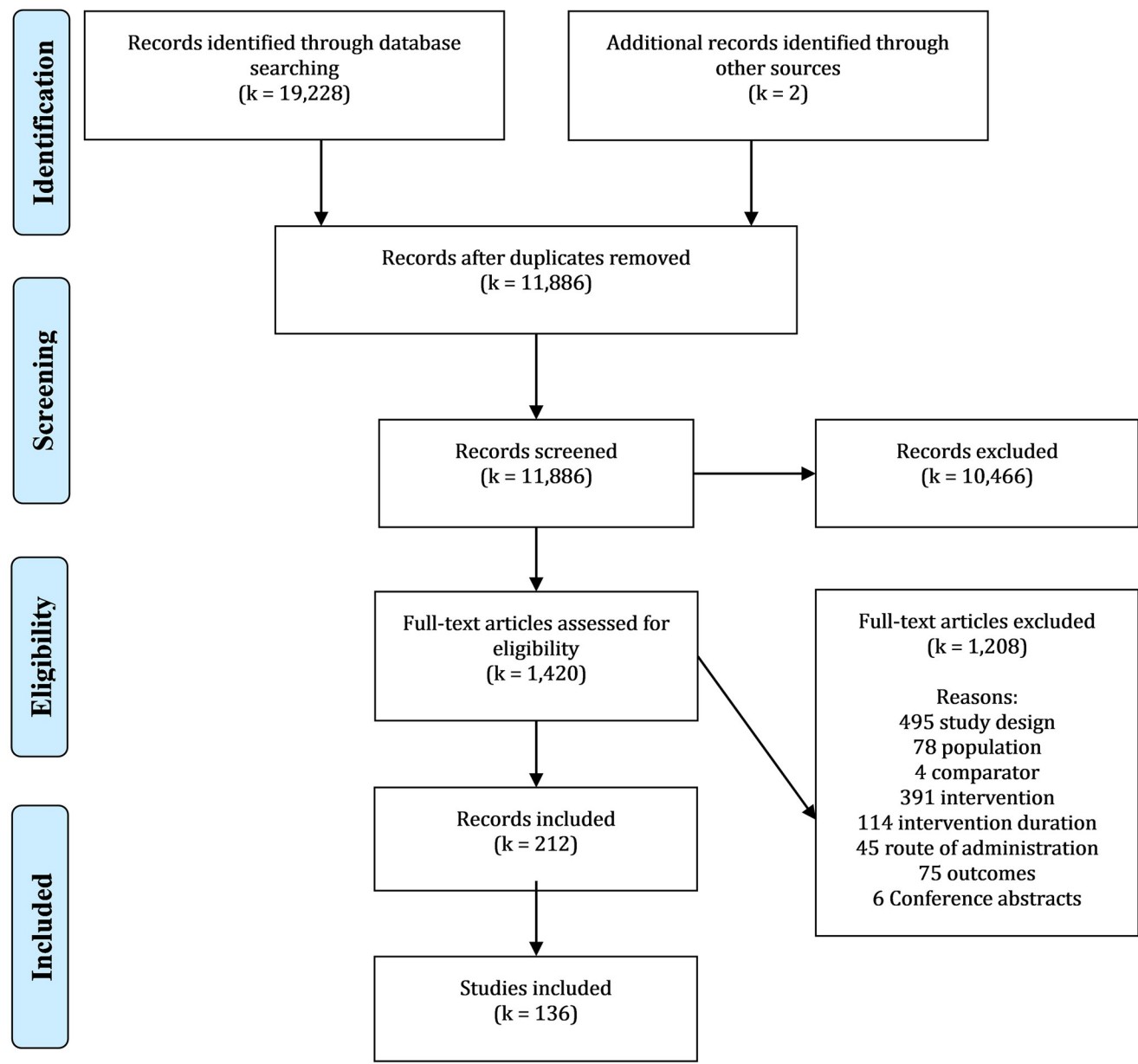

**Fig 1. Study selection flowchart.** *Reasons for full–text exclusion are listed in the order they were assessed.

## Outcomes

As an initial check, we undertook univariate meta–analyses (S1 Appendix). Below, we report primary outcome results of the multivariate meta–analyses; secondary outcomes (i.e., other time point ranges, and cognitive functioning, real life functioning, relationship with the self, and psychosomatic outcomes measured at all time point ranges) are reported in S1 Appendix. When reading this report, outcome improvement or deterioration should be understood as relative to the control group, not to baseline.

In comparison with passive control groups and between 1 and 6 months post–intervention, on average MBPs improved anxiety (SMD = −0.56; 95% CI −0.80 to −0.33; p–value < 0.001;

**Table 1. Characteristics of included studies.**

| First Author, Year, Country | N | Participants (category) | Mean Age (SD) | Women | Outcomes | Intervention (category) | Control/s (category) | Outcome Time Points |
|---|---|---|---|---|---|---|---|---|
| Aeamla–Or 2015 [73], Thailand | 127 | Nursing students (1) | 19 (0.9) | 91% | Dep, Dis, Mindf, Real func, Self | MBSR (3) | No intervention (support as usual) (1) | Post–int, 2 m, 6 m |
| Agee 2009 [74], US | 43 | Community adults (1) | 42 (13) | 91% | Dis, Mindf | Mindfulness meditation (1) | Progressive Muscle Relaxation (3) | Post–int, 1 m, 3 m |
| Allen 2012 [75], Denmark | 61 | Community adults (1) | 27 | 55% | Cog | Mindfulness training (1) | Shared Reading and Learning (2) | Post–int |
| Amutio 2015 [76,77], Spain | 42 | Physicians (2) | 47 (9.4) | 57% | Mindf, Real func | MBSR (3) | Waitlist (1) | Post–int |
| Anclair 2018 [78], Sweden | 21 | Parents of chronically ill children (2) | 41 (6.1) | 93% | Dis, Real func | Here and Now Version 2.0 (2) | CBT intervention (3) | Post–int |
| Anderson 2007 [79], Canada | 86 | Healthy adults (1) | 39 | 92% | Anx, Cog, Dep | MBSR (3) | Waitlist (1) | Post–int |
| Armstrong 2016 [80], UK | 34 | Stressed university students and staff (3) | 30 (8.4) | 91% | Anx, Dep, Mindf, Real func, Self | MBCT (2) | "Get Self–Help" online course (2) | 1 m |
| Arredondo 2017 [81], Spain | 40 | Employees (1) | 37 (5.6) | 78% | Dis, Mindf, Real func, Self | Brief integrated mindfulness practices (2) | Waitlist (1) | Post–int, 3 m |
| Astin 1997 [82], US | 28 | Undergraduate health students (1) | NA | 64% | Dis | MBSR (3) | Waitlist (1) | Post–int |
| Asuero 2014 [83], Spain | 68 | Primary care health professionals (2) | 47 (8) | 92% | Mindf, Real func | MBSR (3) | Waitlist (1) | Post–int |
| Auseron 2018 [84], Spain | 48 | Primary care health professionals (2) | 50 (8.2) | 84% | Dis, Mindf, Real func, Self | Mindfulness and self–compassion (2) | No intervention (1) | Post–int |
| Barrett 2012 [85–91], US | 154 | Older adults (2) | 59 (6.6) | 82% | Dis, WB, Mindf, Somat, Real func | Mindfulness meditation (3) | a) Waitlist (1) b) Exercise program (3) | Post–int, 3 m |
| Barrett 2018 [92–97], US | 413 | Adults aged 30 to 69 years (1) | 50 (11.6) | 76% | Dep, Dis, WB, Mindf, Somat, Real func | MBSR (3) | a) No intervention (1) b) Progressive moderate intensity exercise (3) | Post–int, 2 m, 3 m, 6 m |
| Beattie 2017 [98], Australia | 48 | Pregnant women (2) | 29 | 100% | Dep, Dis, Mindf | Mindfulness in Pregnancy (2) | Pregnancy Support Active Control Intervention (PSP) (3) | Post–int, 1.5 m |
| Behbahani 2018 [99], Iran | 60 | Parents/carers (2) | NA | 100% | Dis | Mindful Parenting Training (1) | No intervention (1) | Post–int, 2 m |
| Benn 2012 [100], US | 70 | Parents and educators of children with special needs (2) | Range 26–60 | 92% | Dep, Dis, WB, Mindf, Self | Stress Management and Relaxation Techniques in Education (2) | Waitlist (1) | Post–int, 2 m |
| Berghmans 2010 [101], France | 26 | Students (1) | 28 (5.8) | 78% | Anx, Dep | MBSR (3) | No intervention (1) | Post–int |
| Black 2015 [102–106], US | 49 | Older adults with sleep disturbance (3) | 66 (7.4) | 67% | Anx, Dep, Dis, Mindf, Somat | MAPs (1) | Sleep Hygiene and Education (3) | Post–int |
| Brown 2016 [107], US | 38 | Carers (2) | 61 (10.4) | 84% | Dis, Real func | MBSR (3) | Near–structurally equivalent to standard Social Support (2) | Post–int, 3 m |
| Carmody 2011 [108], US | 110 | Peri/postmenopausal women (2) | 53 (4.9) | 100% | Anx, Dep, Dis, Mindf, Somat | MBSR (3) | Waitlist (1) | Post–int, 3 m |
| Carson 2004 [109,110], US | 114 | Non–distressed couples (1) | 38 (11.4) | 50% | Dis | Mindfulness–Based Relationship Enhancement (2) | Waitlist (1) | Post–int, 3 m |
| Cerna 2019 [111], Chile | 103 | General adults (1) | 31 | 75% | Dep, WB | Brief mindfulness training program (1) | Waitlist (1) | Post–int |

(*Continued*)

**Table 1.** (Continued)

| First Author, Year, Country | N | Participants (category) | Mean Age (SD) | Women | Outcomes | Intervention (category) | Control/s (category) | Outcome Time Points |
|---|---|---|---|---|---|---|---|---|
| Christopher 2018 [112–114], US | 61 | Law enforcement officers (2) | 44 (6.1) | 13% | Anx, Cog, Dep, Dis, Somat, Real func, Self | Mindfulness–Based Resilience Training (1) | Waitlist (1) | Post–int, 3 m |
| Cohen–Katz 2004 [115,116], US | 27 | Healthcare workers (2) | 46 | 100% | Dis, Mindf, Real func | MBSR (3) | Waitlist (1) | Post–int |
| Corsica 2014 [117], US | 53 | People struggling with emotional eating (3) | 45 (10.4) | 98% | Dis, Real func | a) MBSR merged with b) MBSR plus Stress–eating intervention (3) | Stress–eating intervention (3) | Post–int, 1.5 m |
| Cousin 2016 [118], UK | 87 | Adults (1) | 49 (13) | 77% | Dis, WB | MBCT (2) | Waitlist (1) | Post–int |
| Damião Neto 2019 [119], Brazil | 141 | Medical students (1) | 19 | 50% | Anx, Dep, Dis, Mindf | Mindfulness meditation (1) | "Introduction to University Life" classes (2) | Post–int |
| Davidson 2003 [120], US | 48 | Healthy employees (1) | 36 | 71% | Anx, Dis, WB | MBSR (3) | Waitlist (1) | Post–int, 4 m |
| Delgado 2010 [121], Spain | 36 | Female university students (1) | Range 18–24 | 100% | Anx, Dep, Dis, WB | Mindfulness training (1) | Relaxation training programme (3) | Post–int |
| Delgado–Pastor 2015 [122], Spain | 45 | Female university students (1) | 22 (3.9) | 100% | Dep, Dis, WB, Mindf | a) Mindfulness cognitive training merged with b) Mindfulness interoceptive training group (1) | No intervention (1) | Post–int |
| Desbordes 2012 [123], US | 51 | Healthy adults (1) | 34 (7.7) | 62% | Anx, Dep | Mindful attention training (1) | a) Cognitively–Based Compassion Training (3) merged with b) Health discussion (3) | Post–int |
| De Vibe 2013 [72,124–128], Norway | 293 | Health undergraduate students (1) | 24 (5.2) | 76% | Dis, WB, Mindf, Real func | MBSR (3) | No intervention (1) | Post–int, 2 y, 4 y, 6 y |
| Duncan 2017 [129], US | 29 | First–time pregnant mothers (2) | NA | 100% | Anx, Dep, Dis, Mindf, Somat, Self | Mind in Labour (6) | Standard childbirth education (2) | Post–int |
| Dvorakova 2017 [130], US | 109 | First year undergraduates (1) | 18 (0.4) | 66% | Anx, Dep, WB, Mindf, Somat, Self | Learning to BREATHE (1) | Waitlist (1) | Post–int |
| Dykens 2014 [131], US | 243 | Mothers of disabled children (2) | 41 (8.9) | 100% | Anx, Dep, Dis, WB, Somat | MBSR (3) | Positive Adult Development (3) | Post–int, 1 m, 3 m, 6 m |
| Dziok 2010 [132], US | 52 | Local area adults (1) | 44 (13.4) | 85% | Anx, Dep | Mindfulness meditation (3) | Waitlist (1) | 0–6 m |
| Esch 2017 [133], Germany | 32 | Adults (1) | 27 (7.6) | 67% | Cog, Mindf | Combined breathing/ mindfulness meditation technique (1) | No intervention (1) | Post–int |
| Ferraioli 2013 [134], US | 21 | Parents of disabled children (2) | NA | 67% | Dis, Mindf | Mindfulness–Based Parent Training (2) | Skill–Based Parent Training (2) | Post–int, 3 m |
| Fiocco 2018 [135], Canada | 96 | Older adults (1) | 69 (4.7) | 73% | Dis | Mindfulness–Based Intervention (1) | Reading and Relaxation Program (3) | Post–int |
| Flook 2013 [136], US | 18 | Public elementary school teachers (2) | 43 (9.9) | 89% | Cog, Dis, Mindf, Real func, Self | MBSR (3) | Waitlist (1) | Post–int |
| Frisvold 2009 [137], US | 40 | Stressed midlife female nurses (3) | 48 (5.6) | 100% | Dep, Dis, Mindf, Somat | MBSR (3) | Attention control menopausal education (2) | Post–int, 2 m |
| Galante 2018 [138,139], UK | 670 | University students (1) | Median 22, range 18–53 | 63% | Dis, WB, Real func | Mindfulness Skills for Students (1) | No intervention/waitlist (support as usual) (1) | Post–int, 1–4 m, 10 m |
| Gallego 2014 [140], Spain | 125 | University students (1) | 20 (3.7) | 58% | Dis | Mindfulness group (2) | a) No intervention (1) b) Physical Education (3) | Post–int |

*(Continued)*

**Table 1.** (Continued)

| First Author, Year, Country | N | Participants (category) | Mean Age (SD) | Women | Outcomes | Intervention (category) | Control/s (category) | Outcome Time Points |
|---|---|---|---|---|---|---|---|---|
| Gambrel 2015 [141,142], US | 72 | Pregnant couples (2) | 32 | 52% | Dis, WB, Mindf | Mindful Transition to Parenthood Program (2) | Waitlist (1) | Post–int |
| Giannandrea 2018 [143], Italy | 60 | Adults (1) | 36 (12.1) | 67% | Cog, Mindf | MBSR (3) | Waitlist (1) | Post–int |
| Glass 2019 [144], US | 57 | University athletes (1) | 19 (1.3) | 85% | Anx, Dep, Dis, WB, Mindf | Mindful Sport Performance Enhancement (3) | Waitlist (1) | Post–int |
| Grandpierre 2013 [145], Canada | 40 | University students with academic difficulties (2) | 25 (6.1) | 70% | Cog, Dis, Real func | Mindfulness for Academic Success (2) | Waitlist (1) | Post–int |
| Greenberg 2010 [146,147], Israel | 76 | General adults (1) | 26 (2.5) | 63% | Cog | Mindfulness training (2) | Waitlist (1) | Post–int |
| Greeson 2014 [148], US | 90 | University students (1) | 25 (5.7) | 66% | Dis, Mindf, Somat, Self | Koru (1) | Waitlist (1) | Post–int |
| Guardino 2014 [149], US | 47 | Stressed pregnant women (3) | 33 (4.8) | 100% | Dis, Mindf | MAPs (1) | Reading control group (2) | Post–int, 1.5 m |
| Haarig 2016 [150], Germany | 28 | Adults (1) | 42 (11.8) | 70% | Dep, Mindf, Real func | Mindfulness–Oriented Stress Management Training (1) | Waitlist (1) | Post–int, 3 m |
| Hou 2013 [151,152], China | 141 | Caregivers (2) | 58 (8.8) | 83% | Anx, Dep, Dis, Mindf, Self | MBSR (3) | Self–help health education (2) | Post–int, 3m |
| Huang 2015 [153], Taiwan | 144 | Employees (1) | 43 | 41% | Dis | Mindfulness–based intervention (3) | Waitlist (1) | Post–int, 2 m |
| Hunt 2018 [154], US | 119 | Psychology undergraduates (1) | 19 | 74% | Dep, Dis, WB | a) Mindfulness training merged with b) Multicomponent Mindfulness & Yoga (1) | a) No intervention (1) b) Yoga Alone (3) merged with c) Study Break with a Therapy Dog (3) | Post–int |
| Hwang 2019 [155], Australia | 185 | School teachers (2) | 42 (12.6) | 84% | Dis, Mindf, Somat, Real func, Self | Reconnected (3) | Waitlist (1) | Post–int, 1.5 m |
| Ireland 2017 [156], Australia | 44 | Intern doctors (2) | 27 (4.8) | 64% | Dis, Real func | Mix mindfulness education and practice (2) | 1–hour break per week (2) | Post–int |
| Isbel 2019 [157–159], Australia | 120 | Older adults (1) | 71 | 65% | Dis, Cog, WB | Mindfulness–Based Attention Training Program (1) | Computer–Based Attention Training Program (3) | Post–int |
| Jain 2007 [160], US | 104 | Healthcare students (1) | 25 | 81% | Dis, WB | Mindfulness meditation (3) | a) Waitlist (1) b) Somatic Relaxation (3) | Post–int |
| James 2018 [161], UK | 65 | Students dealing with perfectionism (3) | Range 18–39 | 82% | Anx, Dep, Dis, Mindf, Self | MBCT (2) | Pure Self–Help Intervention (2) | Post–int, 2.5 m |
| Josefsson 2014 [162], Sweden | 98 | Workers (1) | 50 (10.3) | 91% | Anx, Cog, Dep, WB, Mindf | Mindfulness meditation (1) | Relaxation Training Condition (3) | Post–int |
| Kang 2009 [163], South Korea | 41 | Female nursing students (1) | 23 (1.2) | 100% | Dep, Dis | Stress coping program (3) | No intervention (1) | Post–int |
| Kaviani 2008 [164,165], Iran | 30 | Stressed female students (3) | 21.5 | 100% | Anx, Dep | MBCT (2) | Waitlist (1) | Post–int, 1 m, 6 m |
| Kingston 2007 [166], UK | 45 | Students (1) | 23 | 79% | Dis, WB, Mindf | Mindfulness training (1) | Guided visual imagery (3) | Post–int |
| Kirk 2016 [167,168], US | 51 | University staff and students (1) | 32 (10) | 55% | Dis, WB, Mindf | Mindfulness training (1) | Progressive muscle relaxation (3) | Post–int |
| Klatt 2009 [46], US | 48 | Workers (1) | 45 (2.5) | 34% | Dis, Mindf, Somat | MBSR (3) | Waitlist (1) | Post–int |
| Klatt 2016 [169,170], Denmark | 81 | Employees (1) | 43 (9.3) | 69% | Dis, Somat, Real func | Mindfulness in Motion (3) | Waitlist (1) | Post–int |

(*Continued*)

**Table 1.** (Continued)

| First Author, Year, Country | N | Participants (category) | Mean Age (SD) | Women | Outcomes | Intervention (category) | Control/s (category) | Outcome Time Points |
|---|---|---|---|---|---|---|---|---|
| Kor 2019 [171], China (HK) | 36 | Family caregivers (2) | 57 (10.6) | 83% | Anx, Dep, Dis, Real func | Modified MBCT (2) | No intervention (support as usual) (1) | Post-int, 3 m |
| Krick 2019 [172], Germany | 267 | Police officers (2) | 26 (5.6) | 21% | Dis, Mindf, Self | Mindfulness and Resource–Based Worksite Training (2) | No intervention (regular education courses) (1) | Post-int |
| Kuhlmann 2016 [173,174], Germany | 182 | Medical and dental students (1) | 23 (3.9) | 84% | Dis, WB, Mindf, Self | MediMind (2) | a) Waitlist (1) b) Autogenic training (3) | Post-int, 12m |
| Lacerda 2018 [175], Brazil | 77 | Stressed workers (3) | 37 | 57% | Anx, Cog, Dep, Dis, Mindf | PROGRESS (1) | Waitlist (1) | Post-int |
| Lara–Cinisomo 2019 [176,177], US | 23 | Carers of veterans (2) | 58 (12.4) | 96% | Anx, Dep, Dis, Mindf | MBCT (2) | Waitlist (1) | Post-int |
| Lebares 2019 [178,179], US | 21 | Surgery interns (2) | 28 (2.4) | 38% | Cog, Dep, Dis, Mindf, Real func | MBSR (3) | Active control group (3) | Post-int, 10 m |
| Lee 2010 [180], South Korea | 75 | Middle–aged women (1) | 41 (5.8) | 100% | Anx, Dep, WB, Mindf, Somat, Self | MBCT + self–compassion (2) | Waitlist (1) | Post-int |
| Li 2018 [181], China | 34 | General adults (1) | 29 (9.2) | 67% | Cog, Mindf | MBCT (2) | Waitlist (1) | Post-int |
| Lin 2019 [182], China | 110 | Nurses (2) | 32 (6.9) | 93% | Dis, WB, Real func | MBSR (3) | Waitlist (1) | Post-int, 3 m |
| Liu 2013 [183], China | 72 | College and graduate students (1) | 29 (13.4) | 89% | Dis, WB, Mindf | Mindfulness training (2) | Waitlist (1) | Post-int |
| Liu 2015 [184], China | 65 | General adults (1) | 27 (6.7) | 70% | Mindf | MBCT (2) | Waitlist (1) | Post-int |
| Lo 2017 [185], China | 180 | Parents of disabled children (2) | 39 (5.9) | 94% | Dep, Dis, Mindf | Brief Mindfulness–Based Program (2) | Waitlist (1) | Post-int |
| Lonnberg 2020 [186, 187], Sweden | 193 | Pregnant women (2) | 32 | 100% | Dep, Dis, Mindf | Mindfulness–Based Childbirth & Parenting (1) | Lamaze program (3) | Post-int, 3–5 m |
| Lopez–Maya 2019 [188], US | 76 | Stressed adults (3) | 43 (15.3) | 79% | Dep, Dis, Mindf, Self | MAPs (1) | Health Education Program (3) | Post-int |
| Lynch 2018 [189], UK | 38 | University students (1) | 26 (8.3) | 77% | Anx, Dep, Dis, Mindf | Mindfulness–Based Coping with University Life (6) | Waitlist (1) | Post-int |
| Ma 2019 [190], China (HK) | 43 | Stressed adults (3) | 39 (15.1) | 83% | Anx, Cog, Dep, Dis, Mindf | MBCT (2) | Physical exercise program (3) | Post-int, 2 m |
| MacCoon 2012 [191–194], US | 63 | General adults (1) | 48 (10.7) | 63% | Cog, Dis, WB, Mindf, Somat | MBSR (3) | a) Health Enhancement Program (3) b) Waitlist (1) | Post-int, 4 m |
| Malarkey 2013 [169,195], US | 186 | University workers (1) | 50 | 88% | Dep, Dis, Somat | Mindfulness–based intervention (3) | Lifestyle education group (2) | Post-int |
| Malinowski 2017 [196,197], UK | 56 | Older adults (1) | 65 | 73% | Cog, WB, Mindf, Self | Mindfulness training (2) | Brain training group (3) | Post-int |
| Manotas 2014 [198,199], Colombia | 131 | Healthcare workers (2) | 39 (8.2) | 90% | Dis, Mindf | Mindfulness training (3) | Waitlist (1) | Post-int |
| Moody 2013 [200], US and Israel | 47 | Paediatric oncology staff (2) | NA | 80% | Dep, Dis, Real func | Mindfulness–based course (3) | No intervention (1) | Post-int |
| Moritz 2006 [201], Canada | 165 | Stressed individuals (3) | 44 | 78% | Dis | Meditation group (3) | a) Waitlist (1) b) Spirituality teaching program (3) | Post-int, 1 m |

(*Continued*)

**Table 1.** (Continued)

| First Author, Year, Country | N | Participants (category) | Mean Age (SD) | Women | Outcomes | Intervention (category) | Control/s (category) | Outcome Time Points |
|---|---|---|---|---|---|---|---|---|
| Moynihan 2013 [202–208], US | 219 | Older adults (1) | 74 (6.7) | 62% | Cog, Dep, Dis, WB, Mindf, Somat | MBSR (3) | Waitlist (1) | Post–int, 6 m |
| Mrazek 2013 [209], US | 48 | Undergraduate students (1) | 21 (2.1) | 71% | Cog, Real func | Mindfulness class (1) | Nutrition class (2) | Post–int |
| Neece 2014 [210–212], US | 130 | Parents of disabled children (2) | 36 (7.6) | 96% | Dep, Dis, WB | MBSR (3) | Waitlist (1) | Post–int |
| Norouzi 2020 [213], Iran | 40 | Retired athletes (1) | 34 (1.7) | 0% | Anx, Dep, Dis, WB | MBSR (3) | Active control condition (2) | Post–int, 1 m |
| Nyklicek 2008 [214], the Netherlands | 60 | Distressed adults (3) | 46 (9.9) | 67% | Dis, WB | MBSR (3) | Waitlist (1) | Post–int |
| O'Donnell 2017 [215,216], US | 29 | Caregivers (2) | 71 (6.7) | 93% | Dep, Dis, Mindf, Somat, Real func, Self | MBSR (3) | Progressive Muscle Relaxation (3) | Post–int, 2 m, 6 m, ~12 m |
| Oken 2010 [217], US | 31 | Caregivers of relatives with dementia (2) | 65 (9.3) | 81% | Cog, Dep, Dis, Mindf, Somat, Real func, Self | Mindfulness meditation (2) | a) Respite care only (1) b) Dementia education class (3) | Post–int |
| Pan 2018 [218,219], Taiwan | 104 | Pregnant women (2) | 33 (3.8) | 100% | Dep, Mindf, Self | Mindfulness–Based Childbirth and Parenting (2) | Conventional childbirth education (3) | Post–int, 36–week gestation, 3 m after birth |
| Park 2016 [220], South Korea | 60 | Middle–aged women (1) | 54 (5.4) | 100% | Dep, Dis, Somat | Korean MBSR (3) | Waitlist (1) | Post–int |
| Perez–Blasco 2013 [221], Spain | 26 | Breastfeeding mothers (2) | 34 (4.7) | 100% | Dis, WB, Mindf, Self | Mindfulness training (2) | Waitlist (1) | Post–int |
| Perez–Blasco 2016 [222], Spain | 45 | Older adults (1) | 64 (4.1) | 67% | Anx, Dep, Dis, Real func | Mindfulness training (2) | Waitlist (1) | Post–int |
| Phang 2015 [223], Malaysia | 75 | medical students (1) | 21 (1.1) | 76% | Dis, Mindf, Self | Mindful–Gym (3) | Waitlist (1) | Post–int, 6 m |
| Pipe 2009 [224], US | 33 | Nurse leaders (2) | 50 (6.8) | 97% | Dis, Self | Mindfulness meditation (1) | Structured educational series (3) | Post–int |
| Plummer 2018 [225,226], US | 105 | Nursing students (1) | 23 | 93% | Dis, Mindf | Mindfulness–Centred Stress Reduction (1) | No intervention (1) | Post–int, 3m |
| Pots 2014 [227], the Netherlands | 151 | Adults with depressive symptoms (3) | 48 (11.3) | 78% | Anx, Dis, WB, Mindf | MBCT (2) | Waitlist (1) | Post–int |
| Prakash 2015 [228,229], US | 74 | Older adults (1) | 66 (4) | 58% | Cog, Mindf | Mindfulness–Based Attention Training (3) | Lifestyle education (2) | Post–int |
| Richards 2012 [230,231], US | 47 | Undergraduates (1) | 21 (7.5) | 85% | Mindf, Self | Brief mindfulness intervention and LKM exercises (2) | Waitlist (1) | Post–int |
| Richards 2013 [232], US | 30 | Undergraduates (1) | 21 (3.2) | 72% | Mindf, Self | Brief mindfulness intervention (1) | Waitlist (1) | Post–int |
| Robins 2012 [233,234], US | 56 | Adults (1) | 46 (13) | 84% | Mindf, Real func, Self | MBSR (3) | Waitlist (1) | Post–int |
| Roeser 2013 [235–237], US and Canada | 113 | School teachers (2) | 47 (9.2) | 89% | Cog, Dep, Mindf, Real func | Stress Management and Relaxation Techniques in Education (2) | Waitlist (1) | Post–int, 3 m |
| Sampl 2017 [238], Australia | 109 | Undergraduates (1) | 22 (4.6) | 78% | Anx, Dis, Mindf, Real func | Mindfulness–Based Self–Leadership Training (2) | Waitlist (1) | 1–3 m |
| Schellekens 2017 [239–241], the Netherlands | 44 | Lung cancer patient partners (2) | 59 (7.9) | 53% | Dis, Mindf, Self | MBSR (3) | Waitlist (1) | Post–int, 3 m |

(*Continued*)

**Table 1.** (Continued)

| First Author, Year, Country | N | Participants (category) | Mean Age (SD) | Women | Outcomes | Intervention (category) | Control/s (category) | Outcome Time Points |
|---|---|---|---|---|---|---|---|---|
| Schroeder 2018 [242], US | 33 | Primary care physicians (2) | 43 (8.4) | 73% | Dis, Mindf, Real func | Mindful Medicine Curriculum (2) | Waitlist (1) | Post–int, 3 m |
| Sevinc 2018 [243], US | 50 | Adults (1) | 39 (9.6) | 56% | Anx, Dis, Mindf, Self | MBSR (3) | Relaxation response (3) | Post–int |
| Shapiro 1998 [244], US | 78 | Medical students (1) | NA | 56% | Anx, Dis | Stress Reduction and Relaxation Program (3) | Waitlist (1) | Post–int |
| Shapiro 2005 [245], US | 38 | Healthcare professionals (2) | Range 18–65 | NA | Dis, WB, Real func, Self | MBSR (3) | Waitlist (1) | Post–int |
| Shapiro 2019 [246], US | 41 | Medical students (1) | 24 | 78% | Dep, Dis, Mindf | MBSR (3) | Waitlist (1) | Post–int |
| Shearer 2016 [247], US | 74 | Undergraduates (1) | NA | 57% | Dep, Mindf | Mindfulness meditation (3) | a) No intervention (1) b) De–stress with dogs (3) | Post–int |
| Smart 2017 [248,249], Canada | 38 | Healthy older adults (1) | 70 (3.5) | 53% | Anx, Cog, Self | Wisdom Mind (3) | Memory and Aging Program (3) | Post–int |
| ☐tefan 2018 [250], Romania | 71 | Undergraduates (1) | 19 (1) | 93% | Anx, Dep, Dis, Self | MBSR (2) | Waitlist (1) | Post–int |
| Steinberg 2016 [251–254], US | 32 | Intensive Care Unit Personnel (2) | 40 (11.3) | 88% | Dis, Mindf, Real func | Mindfulness in Motion (3) | Waitlist (1) | Post–int |
| Strub 2013 [255], Luxembourg | 20 | Employees (1) | 85% <45 years old | 40% | Dep, Dis, Real func | MBCT (2) | No intervention (1) | Post–int |
| Thomas 2016 [256], United Arab Emirates | 24 | Psychology college students (1) | 21 (2.3) | 76% | Dep | MBSR (3) | Waitlist (1) | Post–int |
| Van Berkel 2014 [257–262], the Netherlands | 257 | Employees (1) | 46 (9.5) | 87% | Mindf, Real func | Mindful Vitality in Practice (2) | No intervention (1) | Post–int, 6 m |
| Van Dam 2014 [263], US | 56 | Stressed adults (3) | 40 (14.4) | 61% | Anx, Dep, Dis, Mindf, Self | Mindfulness meditation (3) | Waitlist (1) | 1 m |
| Van Dijk 2017 [264,265], the Netherlands | 167 | Medical undergraduates (1) | 25 (1.8) | 79% | Dis, WB, Mindf, Real func | MBSR (3) | No intervention (clerkships as usual) (1) | Post–int, 4 m, 9 m, 12 m, 17 m |
| Verweij 2018 [266,267], the Netherlands | 148 | Medical doctors (2) | 31 (4.6) | 88% | WB, Mindf, Real func, Self | MBSR (3) | Waitlist (1) | Post–int |
| Vieten 2008 [268], US | 34 | Pregnant women with mood concerns (3) | 34 (3.8) | 100% | Dep, Dis, Mindf | Mindful Motherhood (2) | Waitlist (1) | Post–int, 1 m |
| Vinesett 2017 [269], US | 21 | Community adults (1) | 48 (8.1) | 100% | Dep, Dis, WB, Real func | MBSR (3) | Ngoma ceremony (3) | Post–int, 1 m |
| Wang 2012 [270], China | 31 | University students (1) | Range 17–25 | 71% | Cog | Mindfulness (2) | Waitlist (1) | Post–int |
| Whitebird 2013 [271,272], US | 78 | Carers (2) | 57 (9.9) | 89% | Dep, Dis, Real func | MBSR (3) | Standard community caregiver education and social support (3) | Post–int, 4 m |
| Williams 2001 [273], US | 138 | Stressed adults (3) | 43 (2.2) | 72% | Dis | MBSR (3) | No intervention (standard educational materials) (1) | Post–int, 3 m |
| Wilson 2012 [274], US | 96 | Working adults (1) | Range 23–64 | 66% | Dis, WB, Mindf, Somat, Real func | (a) MBAP merged with (b) low dose MBSR (5) | No intervention (1) | Post–int, 1 m |
| Wong 2018 [275], China (HK) | 197 | Peri–/postmenopausal women (2) | 52 (3.1) | 100% | Dis, Mindf, Somat | MBSR (3) | Menopause Education Control (3) | Post–int, 3 m, 6 m |
| Woolhouse 2014 [276], Australia | 32 | Pregnant women (2) | 33 (0.6) | 100% | Anx, Dep, Dis, Mindf | MindBabyBody Programme (2) | Care as usual (1) | Post–int |

(*Continued*)

**Table 1.** (Continued)

| First Author, Year, Country | N | Participants (category) | Mean Age (SD) | Women | Outcomes | Intervention (category) | Control/s (category) | Outcome Time Points |
|---|---|---|---|---|---|---|---|---|
| Xu 2015 [277], China | 90 | Adults (1) | 31 (8) | 56% | Dis, Mindf | Mindfulness training (2) | Waitlist (1) | Post–int |
| Yazdanimehr 2016 [278,279], Iran | 80 | Pregnant women (2) | 26 (5.2) | 100% | Anx, Dep, Dis | MiCBT (2) | Usual prenatal care services (1) | Post–int, 1 m |
| Zhang 2018 [280], China | 66 | Pregnant women (2) | 26 (2.6) | 100% | Anx, Dep | MBSR (3) | Prenatal care knowledge as usual (1) | Post–int |

Some studies did not report the mean age and/or its standard deviation. Participant categories according to intervention targeting: (1) Universal; (2) Selective; and (3) Indicated. Intervention categories: (1) no other components; (2) psychoeducation and/or nonmeditative psychological exercises; (3) physical exercises; (4) other types of meditation; (5) arts; and (6) other/unclear. Control/s categories: (1) passive; (2) nonspecific; and (3) specific. Review outcome abbreviations: Anx, Anxiety; Cog, Cognitive functioning; Dep, Depression; Dis, Distress; Mind, Mindfulness; Real func, Real life functioning; Self, Relationship with self; Somat, Psychosomatic outcomes; WB, Mental well–being. Intervention abbreviations: LKM, Loving–Kindness Meditation; MAPs; Mindful Awareness Practices; MBAP, Mindfulness–Based Art Processing; MBCT, Mindfulness–Based Cognitive Therapy; MBSR, Mindfulness–Based Stress Reduction; MiCBT, Mindfulness–Integrated Cognitive Behaviour Therapy. Control treatment abbreviations: CBT, Cognitive Behavioural Therapy. Time point abbreviations: m, Month/s of follow–up post–intervention; Post–int, Post–Intervention; y, Year/s of follow–up post–intervention.

95% PI −1.19 to 0.06), depression (SMD = −0.53; 95% CI −0.72 to −0.34; $p$–value < 0.001; 95% PI −1.14 to 0.07), psychological distress (SMD = −0.45; 95% CI −0.58 to −0.31; $p$–value < 0.001; 95% PI −1.04 to 0.14), and mental well–being (SMD = 0.33; 95% CI 0.11 to 0.54; $p$–value = 0.003; 95% PI −0.29 to 0.94) (Fig 2, S1 Appendix). Effects, according to Cohen's rule of thumb [281], ranged from small (well–being) to moderate (distress, depression, anxiety). However, the prediction intervals indicated that in more than 5% of trial settings, MBPs may not improve anxiety and depression; indeed, in those settings, the outcome scores following MBPs may even be higher for distress and lower for well–being when compared to those following a passive control.

Very few studies compared MBPs with active nonspecific control groups, so results were interpreted with caution (Fig 2, S1 Appendix). On average, MBPs improved depression (SMD = −0.46; 95% CI −0.81 to −0.10; $p$–value = 0.012; 95% PI −1.57 to 0.66) between 1 and 6 months post–intervention with a moderate effect size, although PIs did not rule out other directions of effect. Anxiety showed a trend towards improvement (SMD = −0.47; 95% CI −0.87 to −0.08; $p$–value = 0.019; 95% PI −1.60 to 0.66). There was no evidence to support MBPs improving distress (SMD = −0.14; 95% CI −0.51 to 0.23; $p$–value = 0.47; 95% PI −1.26 to 0.98). Well–being showed improvement (SMD = 1.40; 95% CI 0.35 to 2.46; $p$–value = 0.009; 95% PI −0.19 to 3.00), but only 1 study measured it, so although multivariate meta–analysis "borrows strength" from other outcomes and studies through their correlations, this result is unreliable [282].

Compared with active control interventions designed to deliver specific effects (Fig 2, S1 Appendix), there was no clear evidence that MBPs improved any primary outcome domain (For anxiety: SMD = 0.07; 95% CI −0.20 to 0.35; $p$–value = 0.61; 95% PI −0.34 to 0.48. For depression: SMD = −0.17; 95% CI −0.32 to −0.01; $p$–value = 0.04; 95% PI −0.50 to 0.16. For distress: SMD −0.01; 95% CI −0.15 to 0.13; $p$–value = 0.90; 95% PI −0.33, 0.32. For well–being: SMD = 0.03; 95% CI −0.18, 0.24; $p$–value = 0.79; 95% PI −0.33, 0.39). Too few studies measured anxiety or well–being outcomes for MBPs relative to active control interventions, so their results are unreliable.

Most studies (121, 89%) do not mention having measured adverse events or effects. Of those that did, 12 trials reported no adverse events or effects during the study. One study

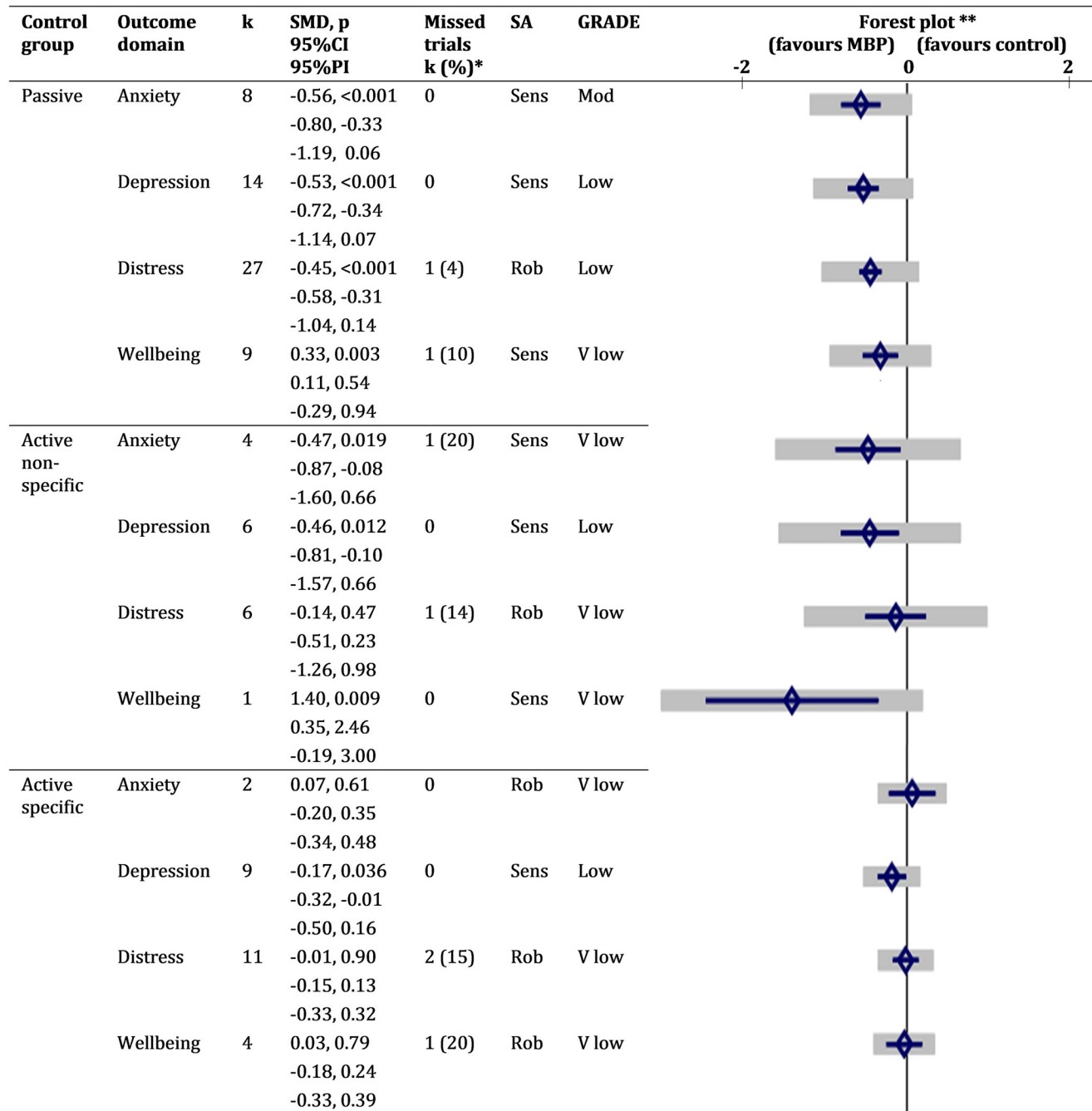

| Control group | Outcome domain | k | SMD, p 95%CI 95%PI | Missed trials k (%)* | SA | GRADE |
|---|---|---|---|---|---|---|
| Passive | Anxiety | 8 | -0.56, <0.001 -0.80, -0.33 -1.19, 0.06 | 0 | Sens | Mod |
| | Depression | 14 | -0.53, <0.001 -0.72, -0.34 -1.14, 0.07 | 0 | Sens | Low |
| | Distress | 27 | -0.45, <0.001 -0.58, -0.31 -1.04, 0.14 | 1 (4) | Rob | Low |
| | Wellbeing | 9 | 0.33, 0.003 0.11, 0.54 -0.29, 0.94 | 1 (10) | Sens | V low |
| Active non-specific | Anxiety | 4 | -0.47, 0.019 -0.87, -0.08 -1.60, 0.66 | 1 (20) | Sens | V low |
| | Depression | 6 | -0.46, 0.012 -0.81, -0.10 -1.57, 0.66 | 0 | Sens | Low |
| | Distress | 6 | -0.14, 0.47 -0.51, 0.23 -1.26, 0.98 | 1 (14) | Rob | V low |
| | Wellbeing | 1 | 1.40, 0.009 0.35, 2.46 -0.19, 3.00 | 0 | Sens | V low |
| Active specific | Anxiety | 2 | 0.07, 0.61 -0.20, 0.35 -0.34, 0.48 | 0 | Rob | V low |
| | Depression | 9 | -0.17, 0.036 -0.32, -0.01 -0.50, 0.16 | 0 | Sens | Low |
| | Distress | 11 | -0.01, 0.90 -0.15, 0.13 -0.33, 0.32 | 2 (15) | Rob | V low |
| | Wellbeing | 4 | 0.03, 0.79 -0.18, 0.24 -0.33, 0.39 | 1 (20) | Rob | V low |

**Fig 2. Summary of primary outcome results (outcome time point is 1–6 months post–intervention follow–up).** *Number of trials with nonreported data for the corresponding outcome. **Diamonds are SMDs, blue bars are 95%CIs, and grey bars are 95% PIs. CI, confidence interval for overall mean; GRADE, Grading of Recommendations Assessment, Development, and Evaluation approach to assess confidence in the cumulative evidence; k, number of trials; MBP, mindfulness–based programme; mod, moderate; PI, prediction interval for new study; Rob, robust; SA, sensitivity analysis; Sens, sensitive; SMD, standardised mean difference; v, very.

reported that some participants "...*experienced adverse emotional, mental or bodily states during mindfulness practice. However, this was not considered to be unintended effects of the intervention, but rather expected results of becoming more mindful of inner experiences*" (page 5) [124]. Two studies reported a participant abandoning the MBP because s/he felt it was being

counterproductive [52,186]. One study actively monitored clinically meaningful adverse events with no significant differences between trial arms [52]. Four studies set up independent data monitoring and ethics committees [52,85,92,171].

## Risk of bias and confidence in the evidence

Fig 3 summarises the risk–of–bias assessments for individual trials (detailed in S1 Appendix). If a study had different outcome–specific ratings for risk of bias from a given source, the highest–risk rating was used in the summary.

All of the included trials are at high risk of bias according to the RoB2, which considers a trial to be at high risk if it scores high for any 1 source of bias. We noted some concerns about biases arising from the randomisation process for 3 quarters of the studies, mainly due to the lack of mention of allocation sequence concealment efforts; the remaining quarter are mostly low risk. Most of the studies were judged to be at high or moderate risk of bias due to deviations from intended interventions. This was mainly due to lack of measurement or description of contamination between trial arms, which is particularly likely in trials with passive control groups where control participants could have potentially learnt elsewhere critical components of MBPs such as mindfulness skills. However, this bias would dampen rather than inflate any effects favouring MBPs, so it is not of major concern.

About 60% of the trials were deemed at high risk of bias due to missing outcome data. The direction of this bias could favour the effects of MBPs because participants who feel unwell may be less likely to attend assessment sessions or complete self–reported outcomes [52]. The high prevalence of the latter accounts for why almost all of the included trials are at high risk of bias in measurement of the outcome, because the assessors are the participants themselves. Very few trials have prospective public protocols that include analysis plans, so for most studies, we noted some concerns as we could not rule out biases in the selection of the reported results.

Regarding selective underreporting or nonreporting of results, Fig 2 (also S1 Appendix) show the number of known nonreported results per outcome domain in the included studies. We also found 6 potentially eligible trial registry records with no available results (S1 Appendix); 3 of them may have measured primary outcomes. To estimate unknown nonreported results, we compiled 3 funnel plots. These revealed evidence of small–study or nonreporting biases in the outcome domain of depression for MBPs compared with passive control groups (S1 Appendix), but not for the distress outcome domain, for MBPs compared with passive (S1 Appendix) and active (S1 Appendix) controls.

Given the overall high risk of bias of the included trials, degree of allegiance to the MBP assessed could play an important role, as suggested in previous studies [283,284]. If we consider that allegiance may be strong where study authors developed and/or taught the MBP, or where relevant conflicts of interest were disclosed, we could rule out allegiance effects (i.e., discard these factors) in only 7 studies (5%).

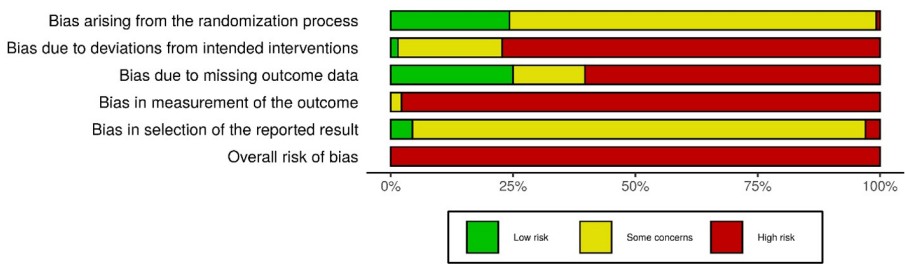

**Fig 3. Risk of bias across studies.** Highest–risk ratings were used for sources with outcome–level assessments.

Fig 2 shows the GRADE assessments for each primary outcome domain (detailed in S1 Appendix). Confidence in the cumulative evidence is low or very low for most outcome domains, except for anxiety in the comparison of MBPs with passive controls, for which we have moderate confidence.

## Sensitivity analyses

To perform a sensitivity analysis of methodological quality, we removed trials deemed to be at high risk of bias from 3 or more sources (most trials have high risk from 2 or 3 sources, so the sample was divided into roughly equal parts). This sensitivity analysis led to reductions in the effects of MBPs on primary outcomes compared to passive controls. The effects on anxiety (SMD = −0.22; 95% CI −0.57 to 0.13; $p$–value = 0.22), depression (SMD = −0.24; 95% CI −0.49 to 0.00; $p$–value = 0.05), and mental well–being (SMD = 0.27; 95% CI 0.02 to 0.58; $p$–value = 0.04) were no longer significant, but the effect on psychological distress was robust (SMD = −0.30; 95% CI −0.48 to −0.11; $p$–value = 0.001) (S1 Appendix). In the comparison of MBPs with active nonspecific controls, the effect on depression was no longer significant (SMD = −0.46; 95% CI −0.90 to −0.02; $p$–value = 0.04), with no changes in the direction or significance of the other outcomes (S1 Appendix). Further details are presented in S1 Appendix.

In the sensitivity analyses testing within–study correlation assumptions by using Riley's method, for MBPs compared with passive controls the effects on well–being lost significance (S1 Appendix), compared with active nonspecific controls the effects on depression lost significance (S1 Appendix), and compared with active specific controls the effects on depression became significant (S1 Appendix). In the sensitivity analyses testing within–study correlation assumptions by conducting univariate meta–analyses, the effect of MBPs compared with passive controls on anxiety lost significance (S1 Appendix), compared with active nonspecific controls the effect on depression lost significance (S1 Appendix), and there was no change in effects compared with specific controls (S1 Appendix).

Primary outcomes were uniformly robust to the few standard deviation imputations made, and to the ICC imputation. The sensitivity analysis of skewed data could only be conducted in the comparison of MBPs with passive controls. There was no change in the size or significance of estimates, but PIs around the effects of MBPs on anxiety, depression, and distress became narrower, excluding adverse scenarios (S1 Appendix).

Several trials reported only the fact that results for some outcomes did not reach statistical significance, rather than the effects themselves. In these cases, we assumed the effect size to be null (i.e., point estimate = 0) and calculated the variance from the sample size. We then conducted post hoc sensitivity analyses setting the effect size to +/−1 standard error. When setting the effect size to +1 standard error, in the comparison of MBPs with active nonspecific controls, the effect on depression lost significance (S1 Appendix); other than that, all of the results were robust to our point estimate imputation (S1 Appendix).

We performed a post hoc sensitivity analysis on our primary outcomes excluding trials with unclear teacher competence. There was no change in the size or significance of the estimates, except for the effect on anxiety in comparison with passive controls, which became stronger and with PIs excluding the null (S1 Appendix).

## Moderator analyses

For the comparison of MBPs with passive controls, the multivariate meta–regression including all time point ranges failed to converge, so we only included the primary range of 1 to 6 months follow–up (S1 Appendix).

For depression, SMDs for MBPs versus passive controls tested in the USA were 1.10 units lower (i.e., less effective) than those tested elsewhere, adjusting for other potential moderators ($p < 0.001$). In a post hoc analysis to explore whether this difference is explained by lower risk of bias in trials conducted in the USA, we included the number of high risk–of–bias sources as a variable in the model: This did not modify the size or significance of this moderation. Running a meta–analysis of non–USA trials only returned a large effect size (SMD −0.93 (95%CI −1.25, −0.62), $p < 0.001$) and narrower prediction intervals that excluded the null effect (−1.66, −0.21). The corollary analysis including only USA–based trials returned a borderline significant small effect (SMD −0.24 (95%CI −0.48, −0.01) $p = 0.04$).

Selective MBPs were 1.10 standard deviations more effective ($p = 0.002$), and indicated MBPs were 0.84 standard deviations more effective ($p = 0.014$; fewer than 5 outcomes in this category) than universal MBPs, in the strength of their benefits compared to passive controls for depression. Running separate analyses for selective and indicated versus universal MBPs did not reduce heterogeneity or significantly modify effects, although the benefits of universal MBPs versus passive controls for depression had a smaller effect size (SMD 0.70 for selective and indicated and 0.39 for universal interventions).

Weaker moderator effects for the depression outcome domain showed that for each extra hour of in–person teaching, the beneficial effect was reduced by 0.05 standard deviations ($p = 0.013$) and that MBPs that included physical exercise were 0.96 standard deviations more effective than MBPs with no additional components ($p = 0.017$, fewer than 5 outcomes in base category) relative to passive controls.

For the effects of MBPs versus passive controls on anxiety, MBPs, which were indicated interventions, were 1.12 standard deviations more effective than universal MBPs, adjusting for other potential moderators ($p = 0.007$, fewer than 5 outcomes in this category). Study location had the same moderating effect as with depression, albeit with borderline significance ($p = 0.028$), which disappeared after adjusting for methodological quality. There were no significant moderator effects for the psychological distress and mental well–being outcomes when comparing MBPs with passive controls.

In comparison with active specific controls, MBPs may be less effective to reduce distress as a selective intervention ($p = 0.02$, S1 Appendix). However, this last analysis (univariate since multivariate meta–analyses failed to converge) only had 11 studies and fewer than 5 studies per category, so the results are unreliable. No other outcome domains could be assessed.

## Discussion

### Summary of findings

We report a systematic review and meta–analysis of RCTs comparing the effects of group–based MBPs delivered in nonclinical settings, versus control conditions, on a range of measures of mental health and functioning.

Our primary outcome results show with a very low to moderate degree of confidence that compared with taking no action (a passive control), MBPs on average improve medium–term mental health outcomes in nonclinical settings. Psychological distress shows the most robust improvement and well–being the smallest improvement, while depression and anxiety show the most homogeneous one.

Compared with taking nonspecific action, MBPs may improve depressive symptoms and the relationship with the self, but reliability is low. Compared with other interventions to improve mental health, we found no indication of MBPs being better or worse.

In general and across comparisons, we cannot be confident that MBPs will confer benefits in every setting. The strongest moderators of MBP effects, which modulated depression and

anxiety outcomes, were study population—MBPs targeted at higher risk populations or at those with subclinical symptoms of mental disorders were more beneficial—and study nationality, with USA–based trials reporting smaller effects than elsewhere. All of the trials included in the review were deemed at high risk of bias, and of all the primary outcomes, only the effects of MBPs relative to passive controls on distress remained when trials with the highest risk of bias were excluded in sensitivity analyses.

## Interpretation and comparison with previous research

Our results present a more complex picture than those of previous reviews, particularly concerning the heterogeneity of effects revealed by wide PIs. PIs show the range of effects to be expected in similar studies to those included in the meta–analysis. In the absence of between–study heterogeneity, PIs equate to CIs (which summarise average effects for the average study); in the presence of such heterogeneity, PIs are wider, meaning that there will be settings where conclusions based on CIs will not hold. Settings encompass a broad range of factors such as type of community, social context, type of MBP, and the way in which the study was conducted; any of these could moderate intervention effects.

Other reviews have also found that their results were sensitive to trial quality [32]. There have been several calls to improve the quality of mindfulness research, and of behavioural interventions research generally, with only modest improvement over time [39,285–287].

Our finding that selective and indicated MBPs were more effective in reducing anxiety and depression than universal MBPs is not unique [29]. It may reflect the fact that those with worse mental health to begin with are more likely to benefit. This finding could also be due to differences in the types of MBPs or their teachers (e.g., those teaching selective or indicated MBPs being therapists). However, the absence of this differential effect on psychological distress outcomes suggests that results could be explained by a ceiling effect: Depression and anxiety questionnaires may be more sensitive to improvement among high–risk or subclinical populations than among those who are less affected, while distress questionnaires may retain sensitivity along this mental health spectrum.

In synthesising studies from different countries and cultures, we tested whether the intervention could have an effect that goes beyond cultural differences. The results obtained, in particular the wide prediction intervals plus the moderation by study location, suggest that cultural and social differences do determine the extent to which MBPs are beneficial [40]. Our moderation analysis tapped into one such difference, as have other recent analyses [49]. Modern mindfulness is an American product undergoing continuous dissemination within the USA since the 1970s [9], so familiarity with it is high. In contrast, a novelty effect, fuelled by advocates in a number of ways (e.g., through researchers' intervention allegiance), may be operating outside of the USA to varying degrees [49]. Also, MBPs may be taught in subtly different ways depending on the culture in which they are modified and delivered.

Little is known about differential effects of various MBP intervention components [288]. In our moderation analyses, we found some support for incorporating physical activity within MBPs. Other effect moderators need to be considered. A recent systematic review of workplace MBPs noted that some individual study effect estimates are opposite to the direction of benefit (see our examples in S1 Appendix), and suggested that not allowing the MBP to take place within working hours could be the cause, since needing extra time to attend the MBP on top of work demands may increase stress [24].

Our weak finding that longer courses may be slightly less beneficial was unexpected, although it could be a result of multiple testing and residual confounding. Other reviews

assessing course duration have not found this to be an important effect moderator [31,32]. Combined, this evidence suggests that MBP courses do not have to be long to be effective.

There is consensus among mindfulness leaders that good teacher training is critical for MBP success [6,289]. However, our preliminary post hoc analysis, in line with previous research, did not find evidence of this factor being influential [290]. Rather than mindfulness credentialing, other related aspects such as type of mindfulness training, teaching and communication skills, or whether the teacher had a similar background to their students, may influence MBP effects.

Whether beneficial effects wear off with longer follow–up periods may depend on continued mindfulness practice, which evidence suggests tails off with time [291]. It may also indicate that a proportion of the effect is nonspecific, including social interaction (by virtue of the group format) and placebo effects, particularly for self–reported outcomes among unblinded participants.

Our results differ in some respects from those of recent reviews looking at MBPs for university students [29,31] and from those looking at MBPs for patients with mental health problems [57,292]. It may be that MBPs are more beneficial to younger populations and to those feeling worse. However, recent reviews of MBPs for children and adolescents have mixed findings [49,56,293]. Contextual factors (e.g., student mindset), intervention characteristics, and review methodologies also need to be considered when comparing results between reviews.

No adverse effects were reported. However, confidence in this result remains low given the low percentage of trials measuring them, as noted before [31,294], and the passive reliance on spontaneous reporting in most studies, which may underestimate adverse effect frequency by more than 20–fold [38,295,296]. The wide prediction intervals found in this review may go some way to explain why unwanted effects are reported in surveys, despite MBPs showing benefit on average [297,298]. It was suggested that unpleasant experiences are part of the intervention effect [124]; it would be important to better understand how common, intense, and heterogeneous these experiences are, both for better intervention targeting and so that commissioners, teachers, and participants know what to expect.

## Strengths and limitations of this review

The strengths of this review include a comprehensive search, detailed prespecification of methods, robust analytic techniques, the fact that none of us have developed or taught any of the included MBPs, and the synthesis of a substantial amount of evidence; these strengths overcome most of the limitations highlighted in an extensive critique of existing healthcare reviews [299]. However, the low quality of most of the primary studies significantly affects confidence and therefore utility of the review results.

MBPs are complex interventions, so quantitative synthesis involved researchers' judgement and simplification [300]. Many different interventions exist, which include the word mindfulness in their title; we carefully selected those that seemed to follow consensus MBP guidelines to obtain meaningful and focused results, but this was not always clear. Some criteria, like including MBPs with a minimum of 4 hours of instruction, or defining the main outcome time point range as between 1 and 6 months post–intervention, were reasoned and predefined, but ultimately arbitrary limits. We made some grouping decisions with undesirably thin data, for example, for control groups and intervention components. The characteristics of the participants in MBPs classed as indicated interventions may overlap with those of participants in clinical settings, although we excluded MBPs which required participants to have a clinical diagnosis. We have not analysed individual–level moderators of effect such as baseline mental health. To address this, we plan to conduct an individual participant data meta–analysis.

### Implications for practice and research

Compared with taking no action, MBPs can be an effective means to promote mental health. But it cannot be expected that MBPs will work in every nonclinical setting. This review showed that MBPs implemented within a wide range of cultures and settings, by different agents, and targeting various groups in the community, can have different effects. The techniques and frameworks taught in MBPs have in turn rich and diverse backgrounds (e.g., early Buddhist psychology, contemplative traditions, cognitive neuroscience, participatory medicine) [6]. The interplays between all these social factors can be expected to exert their own effects over and above any universally human psychophysiological effects.

To understand what happens in which setting, implementation of MBPs should be preceded by or partnered with further studies. This research should be interdisciplinary, involving social scientists to better understand the interplay between complex healthcare interventions, like MBPs, and cultural landscapes [301]. Involving stakeholders in participatory research processes is also likely to shed more light on for whom MBPs may be helpful and in what ways. They could also help intervention developers to adapt MBPs to specific populations considering factors other than teachers' mindfulness training or intervention duration.

In the meantime, it is important for mindfulness practitioners not to assume that MBPs will work universally and to discuss this with their students. It has been shown that MBPs need to be implemented carefully within clinical settings [289]; care is also advised in nonclinical settings, where participants may be more diverse and less supported. In planning MBP provision, those adapted to specific at–risk populations may be a better option than universal MBPs.

The field of online MBPs is growing rapidly both in terms of offer and demand, and the Coronavirus Disease 2019 (COVID–19) pandemic has only accelerated this growth [302]. Meta–analyses suggest that online MBPs may be as effective as their offline counterparts, despite most lacking interactions with teachers and peers [31,47]. If the effects of MBPs vary as widely according to the setting as their offline counterparts, the automatic nature of many online MBPs and their expanded audience raise concerns about the lack of human support. More research comparing effectiveness and safety profiles of different MBP delivery formats head–to–head is needed.

This review suggests that MBPs may have specific effects on common mental health symptoms. However, other preventative interventions may be similarly effective. Apart from effectiveness, other aspects such as cultural acceptability, feasibility, and costs need to be considered when deciding which preventative intervention to implement. Comparative effectiveness research is needed to understand which interventions work best in which setting.

The modest trial quality improvement over time may in part reflect low investment in mental health research [303], and challenges around implementing participant blinding and avoiding outcome self–reporting inherent to behavioural mental health intervention trials. However, it is possible to reduce bias with low–resource measures. Allocation sequence concealment can be done simply, and needs to be reported in publications. Authors could easily encourage participants to complete outcome surveys even when they abandon the MBP and use these data in intention–to–treat analyses. They could also actively ask participants about any unexpected or unwanted effects. It is crucial for future trialists to prospectively register trial protocols in free public registers where they specify a primary outcome measure and time point and include a primary outcome data analysis plan. In their publications, authors need to add more intervention and teacher details, even if it has to be in supplementary materials. More resource–intensive improvements include establishing research teams with no allegiance to the intervention, using active control groups (particularly active nonspecific control groups), and collecting data beyond self–report. Regarding methodological implications for

future reviews, our primary outcome results were sensitive to analytic choices, demonstrating how important it is to publicly prespecify meta–analyses in detail to avoid outcome–led analytic strategy selection.

In sum, compared with taking no action, MBPs promote mental health in the average nonclinical setting but cannot be expected to work in every setting. Although MBPs may have specific effects on some common mental health symptoms, other interventions may be equally effective. MBPs should be implemented with care in nonclinical settings and partnered with well–conducted research.

## Supporting information

**S1 Appendix. Supporting information file.**
(PDF)

**S1 Checklist. PRISMA checklist.**
(PDF)

## Acknowledgments

We are extremely grateful to our professional and public stakeholder groups for their keen involvement. We also thank Joseph Durlak, Alp Arat, Steven Stanley, and Isla Kuhn for their help.

## Author Contributions

**Conceptualization:** Julieta Galante, Tim Dalgleish, Ian R White, Peter B Jones.

**Data curation:** Julieta Galante, Claire Friedrich, Anna F Dawson, Marta Modrego-Alarcón, Pia Gebbing, Tim Dalgleish, Peter B Jones.

**Formal analysis:** Julieta Galante, Claire Friedrich, Anna F Dawson, Marta Modrego-Alarcón, Pia Gebbing, Irene Delgado-Suárez, Radhika Gupta, Lydia Dean, Ian R White.

**Funding acquisition:** Julieta Galante.

**Investigation:** Julieta Galante, Claire Friedrich, Anna F Dawson.

**Methodology:** Julieta Galante, Tim Dalgleish, Ian R White, Peter B Jones.

**Project administration:** Julieta Galante, Claire Friedrich, Anna F Dawson.

**Supervision:** Julieta Galante, Tim Dalgleish, Ian R White, Peter B Jones.

**Visualization:** Julieta Galante, Ian R White.

**Writing – original draft:** Julieta Galante, Claire Friedrich.

**Writing – review & editing:** Julieta Galante, Claire Friedrich, Anna F Dawson, Marta Modrego-Alarcón, Pia Gebbing, Irene Delgado-Suárez, Radhika Gupta, Lydia Dean, Tim Dalgleish, Ian R White, Peter B Jones.

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
