## [Editor Report · Decision Letter 0]

2 Jul 2020

Dear Dr Galante, 

Thank you for submitting your manuscript entitled "Mindfulness-Based Programmes for Mental Health Promotion in Adults in Non-Clinical Settings: A Systematic Review and Meta-Analysis of Randomised Controlled Trials" for consideration by PLOS Medicine.

Your manuscript has now been evaluated by the PLOS Medicine editorial staff, as well as by an academic editor with relevant expertise, and I am writing to let you know that we would like to send your submission out for external peer review.

Kind regards,

Caitlin Moyer, Ph.D.,

Associate Editor

PLOS Medicine

---

## [Decision Letter · Decision Letter 1]

9 Aug 2020

Dear Dr. Galante,

Thank you very much for submitting your manuscript "Mindfulness-Based Programmes for Mental Health Promotion in Adults Beyond Clinical Settings: A Systematic Review and Meta-Analysis of Randomised Controlled Trials" (PMEDICINE-D-20-03011R1) for consideration at PLOS Medicine. 

[LINK]

In light of these reviews, I am afraid that we will not be able to accept the manuscript for publication in the journal in its current form, but we would like to consider a revised version that addresses the reviewers' and editors' comments. Obviously we cannot make any decision about publication until we have seen the revised manuscript and your response, and we plan to seek re-review by one or more of the reviewers. 

We expect to receive your revised manuscript by Aug 28 2020 11:59PM. Please email us (plosmedicine@plos.org) if you have any questions or concerns.

We look forward to receiving your revised manuscript. 

Sincerely,

Emma Veitch, PhD

PLOS Medicine

On behalf of Clare Stone, PhD, Acting Chief Editor,

PLOS Medicine

plosmedicine.org

*Please structure your abstract using the PLOS Medicine headings (Background, Methods and Findings, Conclusions - "Methods and Findings" is a single subsection). 

*In the last sentence of the Abstract Methods and Findings section, please include a brief note about any key limitation(s) of the study's methodology (the authors already note that confidence per GRADE is moderate to very low, this could simply be rephrased slightly to more explicitly state this as a limitation of the SR). 

*At this stage, we ask that you include a short, non-technical Author Summary of your research to make findings accessible to a wide audience that includes both scientists and non-scientists. The Author Summary should immediately follow the Abstract in your revised manuscript. This text is subject to editorial change and should be distinct from the scientific abstract. Please see our author guidelines for more information: https://journals.plos.org/plosmedicine/s/revising-your-manuscript#loc-author-summary

*In the paper the authors state throughout the writeup that they have conducted multivariate meta-analyses, but at the end of the Methods section the authors use the term multivariable instead; can we just check here that the difference in terminology is intended as of course the two terms do have slightly different meanings and often multivariate is used where multivariable would be the correct term.

*One reviewer has queried the statement in the discussion (Limitations section) that the SR authors have a lack of allegiance with regard to MBPs; the academic editor (AE) backed up this concern. Obviously when the researchers are evaluating whether there was allegiance/lack of allegiance in the trials they've included they have some degree of objectivity/independence (except perhaps where they themselves are triallists), but the same is not true of the SR itself. So the AE felt that this statement could be reconsidered and either justified or rephrased in some way, especially since some authors are triallists in some of the included trials. 

Comments from the reviewers:

Reviewer #1: I confine my remarks to statistical aspects of this paper. The general approach is very well done, but I have a couple comments on it and some suggestions for the tables and figures.

NOTE: It would help to have pages and line numbers

Abstract: The sentence about multiple comparisons is unclear. What exactly was done? Bonferonni? Or something else?

 The fact that the GRADE scores were pretty low will reduce the power to find effects, if they are there, and may increase bias.

The first line of the introduction isn't really meaningful. "Top cause" depends on how causes are listed. E.g. is "mental illness" one thing or many? Instead, give some numbers.

Table 2 - I like having as much as posssible of the page devoted to the graph itself. So I would

 1. Put the domain under the group, so it is one column, with domain slightly indented

 2. Make one column with three rows per study that includes row 1 SMD and p value row 2 CI and row 3 PI

 3. Remove the word "confidence"

 4. Expand the graph itself

In addition, there should be a "favors control" label and a scale for the axis.

Figure 2 Stacked bars are not recommended. Maybe a Cleveland dot plot would be good, or maybe a mosaic plot

Peter Flom

Reviewer #2: This is a pre-registered meta-analysis of 136 trials including 11,605 participants examining the effectiveness of Mindfulness Based Programmes to improve psychological distress and wellbeing in non-clinical populations.

Abstract 

For reasons of readability, I'd prefer the authors to refer to the outcome measures in a consistent order throughout the manuscript. There is some variability in how they report this in the abstract and the rest of the manuscript. I'd refer to "interventions without specific effects or outcomes" as "non-specific active control conditions", as is more commonly done. And "specific effect interventions" as "specific active control conditions". I'd also appreciate some specification as to how results differed according to trial and cultural setting in the abstract. It is not entirely clear at this stage what the authors mean by selective and indicated MBPs, so they might need to clarify that as well. And it might be more helpful if the authors specify in which setting they can expect MBPs to work or not, rather than just mention they cannot be expected to work in every setting. 

Introduction

It is a strong point of the meta-analysis that it only includes RCTs, that it includes unpublished reports and studies in other languages than English. And that the review procedures were developed with public and professional stakeholders. 

For reasons of clarity, I would consider to move effects on secondary outcome measures, such as functioning, relationship with the self and psychosomatic symptoms to the supplementary materials. Or even omit altogether from the manuscript.

Methods

The meta-analysis was restricted to group-based first generation MBPs. Particularly with regard to non-clinical populations, online interventions might be both more acceptable and more feasible and are increasingly applied. In this light, I think the authors should comment on their choice in a bit more detail and possibly include this format in their implications for future research or practice. 

As the group-based first generation MBPs classically consist of 8 sessions of 2 hours duration, it is slightly puzzling how the authors could have included interventions of only four one-hour sessions. Could the authors elaborate on this?

The authors have chosen to look at assessments between one and six months follow-up. Does this mean that studies only including post-intervention assessments will be excluded from their main analysis? Shouldn't they report in more detail for how many studies this is the case? Although I can follow their line of reasoning, do the costs of this choice not exceed the benefits? 

The authors consider a trial as being unpublished if enrolment started more than three years before their search date. Given my experience with how long it takes to publish trials like this, is this not a bit too restrictive? 

I'd appreciate if the authors could include a bit more information about which measures were most representative of their primary outcome domains: psychological distress, anxiety, depression and particularly mental wellbeing?

Results

Sixteen trails recruited stressed individuals for whom the MBP was considered an indicated preventative intervention. It is a question whether these individuals should be considered non-clinical or clinical. I think the authors should reflect on this in the discussion section of the manuscript a bit more. Intervention for healthcare workers are considered to be selective, those for students universal. What about medical students, during their internships? 

Opioid use is mentioned as a secondary outcome measure. Is this compatible with a non-clinical population? Opioid abuse is considered to be a psychiatric disorder. 

The authors mention MBPs improved anxiety, depression psychological distress and mental wellbeing. Please use a consistent order in reporting on outcomes for reasons of readability.

I would consider to move the secondary outcomes to supplementary material (post-intervention and longer term outcomes) or even completely omit from the manuscript (cognitive functioning, relationship with the self, etc.). 

Risk of bias: One of the obvious difficulties with offering a psychological intervention is allocation concealment. Does this have a disproportionate effect on the assessment of risk of bias? I think it might be worth reflecting on this in the discussion section of the paper and offering some advice on how authors could go about this. 

I would consider to move the source specific sensitivity analyses to the supplementary materials or remove from the manuscript altogether. 

Did I understand correctly that the number of hours of in-person teaching was associated with a reduction of beneficial effect? This seems to be unexpected, I think the authors should reflect on possible reasons for this in the Discussion section of the paper. 

Discussion

With regard to trial quality, the authors mention the importance to improve the quality of mindfulness research. They do not provide any specific advice about which aspects are most important and how to go about improving them. I think they should include these in this paragraph or in the recommendations for future research: 

- Many trials do not report credentials of MBP teachers.

- Lack of mention of allocation concealment efforts.

- Missing outcome data. 

- Lack of prospective public protocols. 

- Allegiance effects could only be ruled out in seven studies. 

- Very few studies compared MBPs with active non-specific control groups. 

- Most studies do not mention having measured adverse events or effects. 

In terms of cultural differences, the USA are more familiar with MBPs which might be an explanation of them being less effective. Is there any information about possible other sources of differences, such as socio-economic background, education, racial background etc? 

The authors mention that their preliminary post hoc analysis did not find evidence that good teacher training is influential. I seemed to have missed them reporting this in their results section. Could the authors give more details about this? 

Under the strengths of this review the authors mentioned their lack of allegiance. However, some of the authors (including the first author) have previously been involved in mindfulness research, and even in one of the included trials, if I have noticed correctly? 

As previously mentioned, I think the recommendations for research (and practice too) are too general. I think the readers would benefit from more specific advice on how to improve both research and the application of MBPs in non-clinical population based on the findings of this meta-analysis. 

Reviewer #3: The paper addresses an interesting issue in a comprehensive way. It is well written, and the methods are executed impeccably. In fact, their methodological skills in terms of meta-analysis execution seem to be greater than my own, so I will refrain from commenting in this regard.

Some comments:

"Finally, formal meta-analysis of the synthesised data is infrequent and sometimes neglects to disaggregate trials with active versus passive control groups". Although you use "sometimes" maybe you should cite publications that do disaggregate or discard active control groups. For instance, these recent publications: Mindfulness, 10(7), 1193-1216. https://doi.org/10.1007/s12671-018-1062-5 and The Journal of Positive Psychology, 14(5), 625-640. https://doi.org/10.1080/17439760.2018.1519588.

Why you did not include common moderators such as year the study was performed, quality of the study, mean age, or gender proportions of the participants?

The section "Implications for practice and research" seems to be underdeveloped. You put forward some strong ideas and you point the way to new avenues of research, but it is unclear what the impact should be in practice. Should Mindfulness practitioners be careful when working with non-clinical populations (as it has been stressed in some clinical populations)? If yes, in which way? If you are discussing the use of Mindfulness as a preventative intervention why do not you bring forward key literature on the topic (e.g. Preventive Medicine Reports, 5, 150-159. https://doi.org/10.1016/j.pmedr.2016.11.013)?

[LINK]

---

## [Decision Letter · Decision Letter 2]

3 Nov 2020

Dear Dr. Galante,

Thank you very much for re-submitting your manuscript "Mindfulness-Based Programmes for Mental Health Promotion in Adults beyond Clinical Settings: A Systematic Review and Meta-Analysis of Randomised Controlled Trials" (PMEDICINE-D-20-03011R2) for review by PLOS Medicine.

I have discussed the paper with my colleagues and the academic editor and it was also seen again by two reviewers. I am pleased to say that provided the remaining editorial and production issues are dealt with we are planning to accept the paper for publication in the journal.

[LINK]

We look forward to receiving the revised manuscript by Nov 10 2020 11:59PM. 

Sincerely,

Caitlin Moyer, 

Associate Editor 

PLOS Medicine

plosmedicine.org

Requests from Editors:

1.Title: We suggest revising to: “Mindfulness-based programmes for mental health promotion in adults in non-clinical settings: A systematic review and meta-analysis of randomised controlled trials”

2.Data availability statement: At this time your statement reads “All XXX files are available from the XXX database (accession number(s) XXX, XXX.).” Please update the data availability statement with the description of where the data underlying the study may be found.

3. Abstract: Background: In the last sentence of the Abstract’s Background section, please conclude with a clear description of your study’s main question or objective.

4. Abstract: Methods and Findings: How were primary outcomes assessed- anxiety, depression, etc. (for example, the Methods mention that these were psychometrically validated)

5. Abstract: Conclusions: Please revise the first sentence to: “Compared with taking no action, MBPs of the included studies promote mental health in non-clinical settings, but given the heterogeneity between studies, the findings do not support generalization of MBP effects across every setting.” (or similar to clarify the reason for your conclusion regarding generalizability across settings).

6. Author summary: Why was this study done?” Please revise the second bullet point to: “Many randomised controlled trials (RCTs) tested whether mindfulness courses show benefit, but results are varied and, to our knowledge, there are no reviews combining the data from these studies to show an overall effect.”

7. Author Summary: What did the researchers do and find?: Please provide brief mentions of “feel-good practices” as examples for context, and also please combine the third and fourth bullet points: “However, findings from these studies suggest that mindfulness programs are neither better nor worse than other feel-good practices [such as…], and that RCTs in this field tend to be of poor quality, so we cannot be sure that our combined results represent the true effects.”

8. Methods (and throughout text) at line 137, please fix the in-text reference call outs (should be [26,37] and please check all the in-text call outs (remove any extra spaces).

9. Methods: Line 145-146: Thank you for referencing your published protocol. Please be sure to describe here any instances where the analyses reported here differ from the published protocol and when any such changes were made, including any additional analyses or changes in response to reviewer feedback.

10. Methods: Line 148-149: Please include the PRISMA checklist as a supporting information document, and refer to it here (S1_Checklist).

11. Methods: Line 151-152: The search was conducted through January 2020. We request you search be updated to the present time.

12. Methods: Line 225: For clarity, please revise to “None of the authors assessed risk of bias of their own trial”

13. Results: In addition to referring to specific tables, please provide the primary outcome results complete with 95% CIs and p values in the text. Where p values are reported, please note p<0.001 where this is applicable.

14. Results: Line 335: It would be helpful to refer to briefly list the secondary outcomes together with reference to specific files/tables where the various secondary outcome results are reported (e.g. cognitive functioning at post-intervention (supporting information tables S9-11)).

15. Results: Line 337-339: Please report 95% CIs and p values associated with improvements in anxiety, depression, psychological distress and mental wellbeing with MBP vs. passive controls.

16. Results: Line 340-343: Please reference where the statistics are reported underlying the finding that in 5% trial settings, MBPs may have no effect on anxiety/depression and possibly result in higher distress and decreased wellbeing measures.

17. Results: Line 350- 356: Please provide the result with p values/confidence intervals associated with the performance of MBPs in comparison to non-specific active control groups.

18. Results: Line 359: Please revise to clarify “Too few studies measured anxiety or wellbeing outcomes for MBPs relative to active control interventions, so their results are unreliable”

19. Results: Lines 415-421: Please present the statistics, confidence intervals and p values for the results following the exclusion of studies you identified with three or more sources of bias.

20. Results: Line 458: Please change p<0.000 to p<0.001

21. Discussion: Line 602: If possible please use more specific descriptions than “Eastern and Western cultures”.

22. Discussion: Line 576: Please change the heading for this section to read “Strengths and limitations of this review”

23.Discussion: Line 594-596: Where you mention your forthcoming IPD meta-analysis, please revise to: “To address this, We plan to conduct an individual patient meta-analysis, which will allow for the analysis of individual-level moderators of effect, such as baseline mental health."

24. Discussion: Line 617-618: Please provide a citation for “The field of online MBPs is growing rapidly both in terms of offer and demand, and the COVID-19 pandemic has only accelerated this growth.” or remove the reference to COVID-19.

25.Discussion: Please end the section with a one paragraph conclusion, summarizing the main findings and future implications of the analysis.

26.Please remove the sections titled Funding, Declaration of Interests, and Contributors from the text and ensure that this information is accurately entered into the relevant sections of the manuscript submission form.

27. References: Please make sure the reference list uses the "Vancouver" style for reference formatting (capitalization and abbreviations of journal titles, for example), and see our website for other reference guidelines: https://journals.plos.org/plosmedicine/s/submission-guidelines#loc-references

28.Checklist: Please provide the completed PRISMA checklist (and refer to it in the Methods, e.g.: "This study is reported as per the Preferred Reporting Items for Systematic Reviews and Meta-Analyses (PRISMA) guideline (S1 Checklist)." When completing the checklist, please use section and paragraph numbers, rather than page numbers.

29. Table 1. Please provide a brief explanation in the legend as to why standard deviations for age are missing for some studies.

Comments from Reviewers:

Reviewer #1: The authors have addressed my concerns and I now recommend publication. 

Peter Flom

Reviewer #2: I would like to thank the authors for their considerate reactions to the points raised by the reviewers. 

There is one remaining issue that I am still not entirely satisfied with, namely the decision to accept interventions with a minimum of 4 one-hour sessions. The authors refer to previous trials mentioning 4 sessions out of 8 a "minimum dose" (Kuyken 2008; Teasdale, 2000). This, however, was used in the context of differentiating completers versus non-completers of the intervention for the per-protocol analysis. This is entirely different from accepting a minimum of 4 sessions for the whole intervention, from which participants will unevitably miss some in the course of the trial.

However, as this was prespecified in the protocol of the meta-analysis and described as "arbitrary" by the authors in the limitations of the paper, I'll go with this.

[LINK]

---

## [Editor Report · Decision Letter 3]

10 Dec 2020

Dear Dr. Galante,

I am writing concerning your manuscript submitted to PLOS Medicine, entitled “Mindfulness-Based Programmes for Mental Health Promotion in Adults in Non-clinical Settings: A Systematic Review and Meta-Analysis of Randomised Controlled Trials.”

We have now completed our final technical checks and have approved your submission for publication. You will shortly receive a letter of formal acceptance from the editor.

Kind regards,

PLOS Medicine